# Unified Guidance for Geometry-Conditioned Molecular Generation

**Sirine Ayadi**[*,1,2]  **Leon Hetzel**[*,1,2,3]  **Johanna Sommer**[*,1,2]

**Fabian Theis**[1,2,3]  **Stephan Günnemann**[1,2]

[1] School of Computation, Information and Technology, Technical University of Munich
[2] Munich Data Science Institute, Technical University of Munich
[3] Center for Computation Health, Helmholtz Munich
`{si.ayadi, l.hetzel, jm.sommer, f.theis, s.guennemann}@tum.de`

## Abstract

Effectively designing molecular geometries is essential to advancing pharmaceutical innovations, a domain, which has experienced great attention through the success of generative models and, in particular, diffusion models. However, current molecular diffusion models are tailored towards a specific downstream task and lack adaptability. We introduce UniGuide, a framework for controlled geometric guidance of unconditional diffusion models that allows flexible conditioning during inference without the requirement of extra training or networks. We show how applications such as structure-based, fragment-based, and ligand-based drug design are formulated in the UniGuide framework and demonstrate on-par or superior performance compared to specialised models. Offering a more versatile approach, UniGuide has the potential to streamline the development of molecular generative models, allowing them to be readily used in diverse application scenarios.

## 1 Introduction

Diffusion models have emerged as an important class of generative models in various domains, including computer vision [1], signal processing [2], computational chemistry, and drug discovery [3–8]. By gradually adding noise to data samples and learning the reverse process of removing noise, diffusion models effectively transform noisy samples into structured data [9, 10]. In the context of drug discovery, it is essential to effectively address downstream tasks, which often pose specific geometric conditions. Examples of this include (i) Structure-based drug design (SBDD) that aims to create small ligands that fit given receptor binding sites [11], (ii) Fragment-based drug design (FBDD) that designs molecules by elaborating known scaffolds [12, 13], or (iii) Ligand-based drug design (LBDD) which generates molecules that fit a certain shape [14]. Recent works address these tasks by either incorporating specialised models or focusing on conditions that directly resemble molecular structures. In both cases, this narrow focus restricts their adaptability to new or slightly altered settings.

We address the challenge of adaptability by introducing UniGuide, a method that unifies guidance for geometry-conditioned molecular generation, see Fig. 1. The key element for achieving this unification is the *condition map*, which transforms complex geometric conditions to match the diffusion model's

---

*Equal contribution
Project Page: www.cs.cit.tum.de/daml/uniguide

38th Conference on Neural Information Processing Systems (NeurIPS 2024).

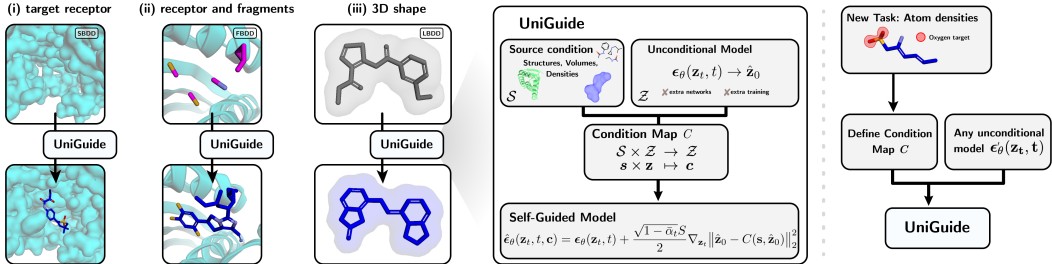

Figure 1: UniGuide handles diverse conditioning modalities for guidance, including: (i) a target receptor for SBDD, (ii) additional molecular fragments for FBDD, or (iii) a predefined 3D shape for LBDD. It combines a source condition $s \in \mathcal{S}$ and the unconditional model $\epsilon_\theta(\mathbf{z}_t, t)$ within its condition map to enable self-guidance. The flexible formulation of our approach can be generalised to new geometric tasks, for example, conditioning on atomic densities.

configuration space, thereby enabling self-guidance without the need for external models. Like other guidance-based approaches, UniGuide does not constrain the generality of the underlying model. Moreover, our method is the most versatile, extending beyond guiding molecular structures to leveraging complex geometric conditions such as volumes, surfaces, and densities, thereby enabling the unified tackling of diverse drug discovery tasks. For complex conditions specifically, previous works primarily rely on conditional diffusion models for effective condition encoding [12–14]. With our method, we are able to tackle the same tasks, while overcoming major drawbacks: UniGuide eliminates the need for additional training and, more importantly, avoids constraining the model to specific tasks.

We demonstrate the wide applicability of UniGuide by tackling a variety of geometry-constrained drug discovery tasks. With performance either on par with or superior to tailored models, we conclude that UniGuide offers advantages beyond its unification. Firstly, while the novelty of conditional models often stems from the condition incorporation, our method redirects focus to advancing unconditional generation, which directly benefits multiple applications. Furthermore, this separation of model training and conditioning allows us to tackle tasks with minimal data, a common scenario in the biological domain.

In summary, our contributions are as follows:

- We present UniGuide: A unified guidance method for generating geometry-conditoned molecular structures, requiring neither additional training nor external networks used to guide the generation.
- We demonstrate UniGuide's wide applicability by tackling various conditioning scenarios in structure-based, fragment-based, and ligand-based drug design.
- We show UniGuide's favourable performance over task-specific baselines, highlighting the practical relevance of our approach.

## 2  Related work

**Diffusion models and controllable generation**   Diffusion models [9, 10] are generative models achieving state-of-the-art performance across various domains, including the generation of images [1, 9], text [15], or point clouds [16]. Conditional diffusion models [17–21] are based on the same principle but incorporate a particular condition in their training, allowing for the controlled generation. Alternatively, classifier guidance [22, 23] relies on external models for controllable generation. Prior works in this context primarily focused on global properties [22, 24], lacking the capacity to condition on the geometric conditions central to our work. For instance, Bao et al. [24] demonstrate control over molecule generation based on desired quantum properties.

***De novo* molecule generation**   Research on *de novo* molecule generation focused extensively on generating molecules using their chemical graph representations [7, 25–34]. However, these methods are limited in modelling the molecules' conformation information and are, therefore, not ideally suited for several drug-discovery settings, such as target-aware drug design. Recently, attention has shifted towards generating molecules in 3D space, utilising variational autoencoders [35], autoregressive models [36–38], flow-based models [39, 40], and diffusion-based approaches [20, 41–47].

**Conditional generation of molecules**  Downstream applications of molecular generation can be categorised by their condition modality. In the case of SBDD [38, 48, 49], Schneuing et al. [11] and Guan et al. [50], for example, introduce models that simultaneously operate on protein pockets and ligands. In the conditional case, the pocket context is fixed throughout the generation. Moreover, FBDD imposes (multiple) scaffolds as a constraint [11, 12, 51–53]. Igashov et al. [13] expand given scaffolds by generating the molecule around the fixed scaffolds. In a related task of FBDD, linker design with pose estimation, as discussed in [54], further generate the rotation of the given scaffolds. SBDD and FBDD rely on the availability of high-quality data of protein pockets, which is often scarce. For this reason, LBDD aims to generate molecules that match the same 3D volume of reference ligands that are known to bind to the target of interest [55, 56]. Chen et al. [14] specifically train a shape encoder to capture the molecular shape of a reference ligand and use the resulting embedding to train a conditional diffusion model.

## 3  Controlling the generation of diffusion models

Diffusion Models [9, 57] learn a Markov Chain that involves a forward process to perturb data from a distribution $q(\mathbf{z})$ and learn to reverse the process to generate new samples from a tractable prior, for example, a normal distribution. Given a data point sampled from the true underlying distribution, $\mathbf{z}_{\text{data}} \sim q(\mathbf{z})$, the forward process $q(\mathbf{z}_t|\mathbf{z}_{t-1})$ gradually adds Gaussian noise:

$$q(\mathbf{z}_t|\mathbf{z}_{t-1}) = \mathcal{N}\big(\mathbf{z}_t \,\big|\, \sqrt{1-\beta_t}\mathbf{z}_{t-1}, \beta_t \mathbf{I}\big) \,, \tag{1}$$

where $\{\beta_t \in (0,1)\}_{t=1}^{T}$ defines a variance schedule. Defining the forward process this way, one can readily sample from $q(\mathbf{z}_t \,|\, \mathbf{z}_{\text{data}})$:

$$\mathbf{z}_t = \sqrt{\bar{\alpha}_t}\mathbf{z}_{\text{data}} + \sqrt{1-\bar{\alpha}_t}\boldsymbol{\epsilon} \,, \quad \boldsymbol{\epsilon} \sim \mathcal{N}(\mathbf{0}, \mathbf{I}) \,, \tag{2}$$

with $\alpha_t = 1 - \beta_t$ and $\bar{\alpha}_t = \prod_{i=1}^{t} \alpha_i$. Since the time-reverse process $q(\mathbf{z}_{t-1}|\mathbf{z}_t)$ depends on $\mathbf{z}_{\text{data}}$, which is not available at generation time, it is approximated by modelling $p_\theta(\mathbf{z}_{t-1} \,|\, \mathbf{z}_t)$:

$$p_\theta(\mathbf{z}_{t-1} \,|\, \mathbf{z}_t) = \mathcal{N}\big(\mathbf{z}_{t-1} \,\big|\, \boldsymbol{\mu}_\theta(\mathbf{z}_t, t), \sigma_t \mathbf{I}\big) \,, \tag{3}$$

where the mean $\boldsymbol{\mu}_\theta$ is parameterised by a noise-predicting neural network $\boldsymbol{\epsilon}_\theta$ in the form of:

$$\boldsymbol{\mu}_\theta(\mathbf{z}_t, t) = \frac{1}{\sqrt{\alpha_t}}\Big(\mathbf{z}_t - \frac{\beta_t}{\sqrt{1-\bar{\alpha}_t}}\boldsymbol{\epsilon}_\theta(\mathbf{z}_t, t)\Big) \,. \tag{4}$$

The model $\boldsymbol{\epsilon}_\theta$ is trained to optimise the variational lower bound through the simplified training objective:

$$\mathcal{L}_{\text{train}} = \frac{1}{2}\big\|\boldsymbol{\epsilon} - \boldsymbol{\epsilon}_\theta(\mathbf{z}_t, t)\big\|_2^2 \quad . \tag{5}$$

**Self-guiding diffusion models**  Using Bayes' rule, the conditional probability $p_\theta(\mathbf{z}_t \,|\, \boldsymbol{c})$ given a condition $\boldsymbol{c}$ can be expressed as

$$p_\theta(\mathbf{z}_t \,|\, \boldsymbol{c}) \propto p_\theta(\mathbf{z}_t) \, p_\theta(\boldsymbol{c} \,|\, \mathbf{z}_t) \,. \tag{6}$$

This allows us to decompose the score function as follows:

$$\nabla_{\mathbf{z}_t} \log p_\theta(\mathbf{z}_t \,|\, \boldsymbol{c}) = \nabla_{\mathbf{z}_t} \log p_\theta(\mathbf{z}_t) + S \, \nabla_{\mathbf{z}_t} \log p_\theta(\boldsymbol{c} \,|\, \mathbf{z}_t) \,, \tag{7}$$

where the second term is used for guiding the unconditional generation, with $S > 0$ controlling the guidance strength. Using that $\nabla_{\mathbf{z}_t} \log p_\theta(\mathbf{z}_t) = -(1-\bar{\alpha}_t)^{-\frac{1}{2}}\boldsymbol{\epsilon}_\theta(\mathbf{z}_t, t)$ [22], we can rewrite the score function from Eq. (7) and identify the modified noise predictor $\hat{\boldsymbol{\epsilon}}_\theta$:

$$\nabla_{\mathbf{z}_t} \log p_\theta(\mathbf{z}_t \,|\, \boldsymbol{c}) = -\frac{1}{\sqrt{1-\bar{\alpha}_t}}\Big[\underbrace{\boldsymbol{\epsilon}_\theta(\mathbf{z}_t, t) - \sqrt{1-\bar{\alpha}_t}S \, \nabla_{\mathbf{z}_t} \log p_\theta(\boldsymbol{c} \,|\, \mathbf{z}_t)}_{=:\hat{\boldsymbol{\epsilon}}_\theta(\mathbf{z}_t, t, \boldsymbol{c})}\Big] \tag{8}$$

The modified mean function $\hat{\boldsymbol{\mu}}_\theta$ then follows from the modified version of Eq. (4), enabling us to sample from $p_\theta(\mathbf{z}_{t-1} \,|\, \mathbf{z}_t, \boldsymbol{c}) \sim \mathcal{N}\big(\hat{\boldsymbol{\mu}}_\theta(\mathbf{z}_t, t, \boldsymbol{c}), \sigma_t \mathbf{I}\big)$:

$$\hat{\boldsymbol{\mu}}_\theta(\mathbf{z}_t, t, \boldsymbol{c}) = \frac{1}{\sqrt{\alpha_t}}\Big(\mathbf{z}_t - \frac{\beta_t}{\sqrt{1-\bar{\alpha}_t}}\hat{\boldsymbol{\epsilon}}_\theta(\mathbf{z}_t, t, \boldsymbol{c})\Big) = \boldsymbol{\mu}_\theta(\mathbf{z}_t, t) + \lambda(t)\nabla_{\mathbf{z}_t} \log p_\theta(\boldsymbol{c} \,|\, \mathbf{z}_t) \,, \tag{9}$$

where $\lambda(t) = (\alpha_t)^{-\frac{1}{2}} \beta_t S$ balances the conditional update. Eq. (9) requires sampling from $\log p_\theta(\boldsymbol{c} \,|\, \mathbf{z}_t)$ to which we do not have access. Assuming the condition $\boldsymbol{c}$ lies in the same space as $\mathbf{z}_t$, we can follow Kollovieh et al. [58] and approximate $\log p_\theta(\boldsymbol{c} \,|\, \mathbf{z}_t)$ as a multivariate Gaussian distribution:

$$p_\theta(\boldsymbol{c} \,|\, \mathbf{z}_t) = \mathcal{N}\big(\boldsymbol{c} \,|\, \boldsymbol{f}_\theta(\mathbf{z}_t, t), \boldsymbol{I}\big) \,, \tag{10}$$

where $\boldsymbol{f}_\theta(\mathbf{z}_t, t)$ approximates the clean data point, enabling to estimate the condition in data space. Using Eq. (2), we can readily predict the clean data point given the noisy sample $\mathbf{z}_t$ via

$$\boldsymbol{f}_\theta(\mathbf{z}_t, t) = \frac{\mathbf{z}_t - \sqrt{1 - \bar{\alpha}_t}\, \boldsymbol{\epsilon}_\theta(\mathbf{z}_t, t)}{\sqrt{\bar{\alpha}_t}} =: \hat{\mathbf{z}}_0 \quad . \tag{11}$$

With this, the *guiding term* becomes a direct differentiation of the squared error with respect to the noisy sample $\mathbf{z}_t$:

$$\nabla_{\mathbf{z}_t} \log p_\theta(\boldsymbol{c} \,|\, \mathbf{z}_t) = -\frac{1}{2} \nabla_{\mathbf{z}_t} \big\| \boldsymbol{f}_\theta(\mathbf{z}_t, t) - \boldsymbol{c} \big\|_2^2 \quad . \tag{12}$$

By directly leveraging the prediction of the unconditional model $\boldsymbol{\epsilon}_\theta$, Eq. (12) establishes our self-guiding conditioning, thereby defining the self-guided noise predictor $\hat{\boldsymbol{\epsilon}}_\theta$:

$$\hat{\boldsymbol{\epsilon}}_\theta(\mathbf{z}_t, t, \boldsymbol{c}) = \boldsymbol{\epsilon}_\theta(\mathbf{z}_t, t) + \frac{\sqrt{1 - \bar{\alpha}_t}\, S}{2} \nabla_{\mathbf{z}_t} \big\| \hat{\mathbf{z}}_0 - \boldsymbol{c} \big\|_2^2 \quad . \tag{13}$$

## 4  UniGuide

To enable the application of unconditional molecular diffusion models $\boldsymbol{\epsilon}_\theta$ to geometric downstream tasks in drug discovery, we aim to develop a unified guidance framework, UniGuide, see Fig. 1. Importantly, we seek to enable guidance from arbitrary geometric conditions $\boldsymbol{s} \in \mathcal{S}$, where $\mathcal{S}$ denotes a general space of source conditions. However, the source conditions $\boldsymbol{s}$ cannot be directly used for the loss computation in Eq. (12) when they do not match the configuration space $\mathcal{Z}$.

To address this challenge, we introduce *condition maps* $C$, which bridge the gap between arbitrary source conditions $\boldsymbol{s}$ and target conditions $\boldsymbol{c}$ suitable for guidance. In Sec. 4.1, we start with its general formulation and continue to derive a condition map $C_\mathcal{Z}$ for the special case where $\mathcal{S} = \mathcal{Z}$. This will be useful when discussing the application of UniGuide to various drug discovery tasks in Sec. 4.2. We also demonstrate how to derive a task-specific condition map $C_{\partial V}$ for ligand-based drug design.

**Notation**   In 3D space, the *configuration* of molecules, including proteins, can be represented by a set of tuples $\mathbf{z} = \{(\boldsymbol{x}_i, \boldsymbol{h}_i)\}_{i=1}^N \in \mathcal{Z}$, where $\boldsymbol{x}_i \in \mathbb{R}^3$ and $\boldsymbol{h}_i \in \mathbb{R}^d$ refer to coordinates and features of a node $\boldsymbol{z}_i = (\boldsymbol{x}_i, \boldsymbol{h}_i)$, respectively. The space of configurations is denoted by $\mathcal{Z}$ and includes configurations of varying size $N$. We distinguish between different configuration entities via superscripts, i.e. refer to molecules $\mathcal{M}$ and proteins $\mathcal{P}$ through $\mathbf{z}^\mathcal{M}$ and $\mathbf{z}^\mathcal{P}$, respectively. The collection of coordinates $\mathbf{x} = \{\boldsymbol{x}_1, \dots, \boldsymbol{x}_N\} \in \mathbb{R}^{N \times 3} \in \mathcal{X}$ defines the *conformation* of a molecule $\mathcal{M}$ or protein $\mathcal{P}$. We represent arbitrary geometric conditions with the variable $\boldsymbol{s} \in \mathcal{S}$, and conditions that can be used for guidance with the variables $\boldsymbol{c} \in \mathcal{Z}$.

### 4.1  Unified self-guidance from geometric conditions $s \in \mathcal{S}$

The concept of a condition map $C$ is essential to our method, enabling guidance from conditions $\boldsymbol{s} \in \mathcal{S}$ in a unified fashion, where $\mathcal{S}$ represents a space of general geometric objects such as structures, densities, or surfaces. These geometric objects do not necessarily match the configuration space $\mathcal{Z}$, i.e. $\mathcal{S} \neq \mathcal{Z}$, preventing the computation of the guiding score function from Eq. (12). We overcome this challenge by defining $C$ as a transformation that maps $\boldsymbol{s}$ to a suitable target condition $\boldsymbol{c} \in \mathcal{Z}$, which is then utilised for self-guidance.

In the most general case, $C$ takes the form of

$$\begin{aligned} C : \mathcal{S} \times \mathcal{Z} &\to \mathcal{Z} \\ \boldsymbol{s} \times \mathbf{z} &\mapsto \boldsymbol{c} \quad , \end{aligned} \tag{14}$$

where the source condition $\boldsymbol{s}$ together with a configuration $\mathbf{z}$ are mapped to a target condition $\boldsymbol{c} \in \mathcal{Z}$. Including the condition map $C$ in the guidance, we obtain our guidance signal:

$$\nabla_{\mathbf{z}_t} \log p_\theta(\boldsymbol{c} \,|\, \mathbf{z}_t) = -\frac{1}{2} \nabla_{\mathbf{z}_t} \big\| \hat{\mathbf{z}}_0 - C(\boldsymbol{s}, \hat{\mathbf{z}}_0) \big\|_2^2 = -\nabla_{\mathbf{z}_t} \mathcal{L}(\hat{\mathbf{z}}_0, \boldsymbol{s}) \quad , \tag{15}$$

where $\hat{\mathbf{z}}_0 = \boldsymbol{f}_\theta(\mathbf{z}_t, t)$ is the estimate of $\mathbf{z}_0$ given the unconditional model $\boldsymbol{\epsilon}_\theta(\mathbf{z}_t, t)$ obtained according to Eq. (11) and $\boldsymbol{c} = C(\boldsymbol{s}, \hat{\mathbf{z}}_0)$ is the target condition produced by the condition map. In this formulation, $\boldsymbol{c}$ can also be understood as *guidance target* of the unconditional model.

It is important to highlight that Eq. (15) should not destroy the underlying properties of the unconditional generative process. In particular, if the unconditional model $\boldsymbol{\epsilon}_\theta$ is equivariant to a set of transformations $G$, e.g. rotations and translations, as is common in the molecular domain, we want to retain equivariance also in the guidance signal. Hence, the self-guided model $\hat{\boldsymbol{\epsilon}}_\theta$ should satisfy

$$\hat{\boldsymbol{\epsilon}}_\theta\big(G(\mathbf{z}_t), t, \boldsymbol{c}\big) = G\big(\hat{\boldsymbol{\epsilon}}_\theta(\mathbf{z}_t, t, \boldsymbol{c})\big) , \tag{16}$$

for all transformations $G$ to which $\boldsymbol{\epsilon}_\theta$ is equivariant.

**Theorem 4.1.** *Consider a function $C : \mathcal{S} \times \mathcal{Z} \to \mathcal{Z}$. If $C(\boldsymbol{s}, \mathbf{z})$ is invariant to rigid transformations $G$ in the first argument and equivariant in the second argument, then the gradient $\nabla_{\mathbf{z}} \|\boldsymbol{v}\|_2^2$ of the vector $\boldsymbol{v} = \mathbf{z} - C(\boldsymbol{s}, \mathbf{z})$ is equivariant to transformations of $\mathbf{z}$.*

*Proof.* We prove Theorem 4.1 in App. B. □

Using Theorem 4.1, we can guarantee equivariant guidance signals if the condition maps $C(\boldsymbol{s}, \mathbf{z})$ are invariant and equivariant under rigid transformations concerning the source condition $\boldsymbol{s}$ and configuration $\mathbf{z}$, respectively.

**Guidance in the special case of $\mathcal{S} = \mathcal{Z}$** In the case where the source condition $\boldsymbol{s}$ directly defines subset $\mathcal{A}$ of $m < N$ nodes of the configuration, i.e. $\mathcal{S} = \mathcal{Z}$, we can fully specify the condition map. This is feasible because the condition map no longer needs to bridge different spaces; it only needs to ensure equivariance, as the loss computation between $\boldsymbol{s}$ and the configuration is already possible. To distinguish this special case from the general setting, we denote $\boldsymbol{s} = \tilde{\mathbf{z}} \in \mathbb{R}^{m \times (3+d)}$ and refer to the defined subset within the configuration $\hat{\mathbf{z}}_0$ by $\hat{\mathbf{z}}_0^{\mathcal{A}}$.

In order to satisfy the requirements on $C\big(\tilde{\mathbf{z}}, \hat{\mathbf{z}}_0^{\mathcal{A}}\big)$ as stated by Theorem 4.1, we align $\tilde{\mathbf{z}}$ with $\hat{\mathbf{z}}_0^{\mathcal{A}}$ by using the Kabsch algorithm [59, 60]. Denoting the resulting transformation with $T_{\hat{\mathbf{z}}_0^{\mathcal{A}}}$, we get an $\hat{\mathbf{z}}_0$-equivariant condition map:

$$\begin{aligned} C_{\mathcal{Z}}: \quad \mathbb{R}^{m \times (3+d)} &\times \mathbb{R}^{m \times (3+d)} \to \mathbb{R}^{m \times (3+d)} \\ \tilde{\mathbf{z}} &\times \hat{\mathbf{z}}_0^{\mathcal{A}} \quad\quad \mapsto \quad T_{\hat{\mathbf{z}}_0^{\mathcal{A}}} \tilde{\mathbf{z}} \quad . \end{aligned} \tag{17}$$

Taken together, we can compute the guidance signal based on the following loss $\mathcal{L}$:

$$\mathcal{L}\big(\hat{\mathbf{z}}_0^{\mathcal{A}}, \tilde{\mathbf{z}}\big) = \frac{1}{2} \big\| \hat{\mathbf{z}}_0^{\mathcal{A}} - T_{\hat{\mathbf{z}}_0^{\mathcal{A}}} \tilde{\mathbf{z}} \big\|_2^2 \quad . \tag{18}$$

We emphasise that although the loss $\mathcal{L}\big(\hat{\mathbf{z}}_0^{\mathcal{A}}, \tilde{\mathbf{z}}\big)$ is computed on the subset $\mathcal{A}$, the gradient, as presented in Eq. (15), is still computed with respect the full configuration $\mathbf{z}_t$.

In summary, our method requires only an unconditionally trained model $\boldsymbol{\epsilon}_\theta$ and a suitable condition map $C$, eliminating the need for additional networks or training. Together, this facilitates unified self-guidance from arbitrary geometric sources. Importantly, the separation of model training and conditioning enables us to tackle tasks even with minimal data, which is crucial in practical scenarios. In the following section, we discuss the wide applicability of UniGuide by illustrating its application to multiple drug discovery tasks.

## 4.2 UniGuide for drug discovery

Having introduced both the guidance framework and the condition map, we will continue to discuss how to tackle a set of drug discovery tasks within the UniGuide framework. We start with its application to ligand-based drug design (LBDD), which aims to generate a ligand that satisfies a predefined molecular shape.

**Ligand-based drug design** LBDD aims to generate novel ligands with a similar 3D shape as a reference ligand $\mathcal{M}_{\text{ref}}$. In this setting, one operates on the molecule level only since the protein information is assumed to be unknown. However, to still generate active ligands that bind to a protein pocket, one leverages the 3D shape information of a reference molecule. Specifically, the goal is to modify the generative process $\hat{\epsilon}_\theta$ to generate a ligand $\mathbf{z}_0$ with a similar 3D shape but different molecular structure than $\mathcal{M}_{\text{ref}}$. With Sec. 4.1 introducing all required concepts, we can readily formulate a *surface condition map* $C_{\partial V}$ suitable to tackle the task of LBDD, see Fig. 2:

To represent $\mathcal{M}_{\text{ref}}$'s 3D shape, we identify our source condition $\boldsymbol{s}$ with a set of $K$ points $\mathbf{y}$ sampled uniformly from the reference ligand's surface $\partial V$, $\mathbf{y} \in \mathbb{R}^{K \times 3} = \mathcal{S}$. As no features are guided, we formulate $C_{\partial V}$ with respect to the conformation space $\mathcal{X} = \mathbb{R}^{N \times 3}$:

$$
\begin{aligned}
C_{\partial V} : \mathbb{R}^{K \times 3} \times \mathbb{R}^{N \times 3} &\to \mathbb{R}^{N \times 3} \\
\mathbf{y} \times \hat{\mathbf{x}}_0 &\mapsto \boldsymbol{c_x} \quad,
\end{aligned}
\tag{19}
$$

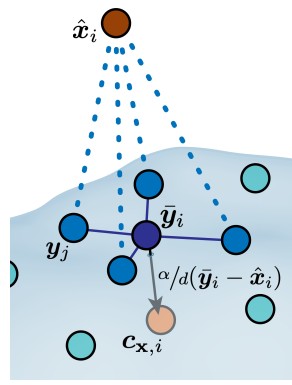

Figure 2: Surface condition map $C_{\partial V}$: For each atom coordinate $\boldsymbol{x}_i$, the closest surface points $\boldsymbol{y}_j$ are computed. The target condition $\boldsymbol{c}_{\mathbf{x},i}$ is the projection along the mean of neighbours $\bar{\boldsymbol{y}}_i$ to the inside of the volume by a margin $\alpha$, where $d = \|\bar{\boldsymbol{y}}_i - \hat{\boldsymbol{x}}_i\|_2$.

where $\hat{\mathbf{x}}_0$ denotes the conformation of the clean data point estimation $\hat{\mathbf{z}}_0$ as computed by Eq. (11). To satisfy Theorem 4.1, $C_{\partial V}$ first aligns $\mathbf{y}$ with $\hat{\mathbf{x}}_0$ by a rotation $R_{\hat{\mathbf{x}}_0} \in \mathbb{R}^{3 \times 3}$ resulting from the ICP algorithm [61]. For every atom coordinate $\hat{\boldsymbol{x}}_i$, $C_{\partial V}$ subsequently computes the mean $\bar{\boldsymbol{y}}_i$ over $\hat{\boldsymbol{x}}_i$'s $k$ closest surface points:

$$
\bar{\boldsymbol{y}}_i = \frac{1}{k} \sum_{j \in \mathcal{N}_{\hat{\boldsymbol{x}}_i}} R_{\hat{\mathbf{x}}_0} \boldsymbol{y}_j \quad, \text{ with } \quad \mathcal{N}_{\hat{\boldsymbol{x}}_i} = \arg \min_{I \subset \{1,\dots,K\}, |I| = k} \sum_{j \in I} \left\| R_{\hat{\mathbf{x}}_0} \boldsymbol{y}_j - \hat{\boldsymbol{x}}_i \right\|_2 \quad.
\tag{20}
$$

Finally, the individual components $\boldsymbol{c}_{\mathbf{x},i}$ of the target condition compute as follows:

$$
\boldsymbol{c}_{\mathbf{x},i} = \begin{cases}
\bar{\boldsymbol{y}}_i + \frac{\alpha}{d}(\bar{\boldsymbol{y}}_i - \hat{\boldsymbol{x}}_i), & \text{if } \hat{\boldsymbol{x}}_i \text{ outside } V \\
\bar{\boldsymbol{y}}_i - \frac{\alpha}{d}(\bar{\boldsymbol{y}}_i - \hat{\boldsymbol{x}}_i), & \text{if } \hat{\boldsymbol{x}}_i \text{ inside } V \wedge d < \alpha \\
\hat{\boldsymbol{x}}_i, & \text{otherwise,}
\end{cases}
\tag{21}
$$

where $d$ denotes the distance to the surface, $d = \|\bar{\boldsymbol{y}}_i - \hat{\boldsymbol{x}}_i\|_2$, and $\alpha$ the required distance to the surface. Note that the target condition $\boldsymbol{c_x}$ represents a valid conformation inside the surface $\partial V$, and that $C_{\partial V}$ effectively bridges spaces from $\mathcal{S}$ to $\mathcal{X}$. Consequently, when using $C_{\partial V}$, the guidance signal is derived from Eq. (15) with the loss function $\mathcal{L}(\hat{\mathbf{z}}_0, \mathbf{y})$. The full algorithm for guidance using $C_{\partial V}$ is presented in App. D.1.

**Structure-based drug design** The goal of SBDD is to design a ligand that binds to a target protein pocket $\boldsymbol{s}$. In this setting, one operates on both the molecule and protein level. Technically, we are interested in generating a ligand $\mathbf{z}_0^{\mathcal{M}}$ conditioned on the protein configuration $\tilde{\mathbf{z}}^{\mathcal{P}}$. With the unconditional diffusion model $\epsilon_\theta(\mathbf{z}_t, t)$, $\mathbf{z}_t = (\mathbf{z}_t^{\mathcal{M}}, \mathbf{z}_t^{\mathcal{P}})$, approximating the joint distribution of ligand-protein pairs $p(\mathbf{z}_{\text{data}}^{\mathcal{M}}, \mathbf{z}_{\text{data}}^{\mathcal{P}})$, one can readily see that the source condition directly corresponds to the configuration of the protein pocket. Hence, we can use $C_{\mathcal{Z}}$ from Sec. 4.1 and identify $\tilde{\mathbf{z}}$ with $\tilde{\mathbf{z}}^{\mathcal{P}}$. The guidance signal then follows from the loss $\mathcal{L}(\hat{\mathbf{z}}_0^{\mathcal{P}}, \tilde{\mathbf{z}}^{\mathcal{P}})$ with $\boldsymbol{c}^{\mathcal{P}} = C_{\mathcal{Z}}(\tilde{\mathbf{z}}^{\mathcal{P}}, \hat{\mathbf{z}}_0^{\mathcal{P}})$ as defined in Eq. (18). We describe the sampling algorithm for the SBDD task in App. E.1.

**Fragment-based drug design** FBDD aims to design a ligand by optimising a molecule around fragments $\mathcal{F}$ that bind weakly to a receptor. Similarly to SBDD, one operates on both the molecule and protein level. Technically, we are interested in generating a ligand $\mathbf{z}_0^{\mathcal{M}}$ conditioned on both the protein and the fragment configuration, $\tilde{\mathbf{z}}^{\mathcal{P}}$ and $\tilde{\mathbf{z}}^{\mathcal{F}}$, respectively. Considering the same kind of unconditional model $\epsilon_\theta(\mathbf{z}_t, t)$ as in SBDD, we can use $C_{\mathcal{Z}}$ from Sec. 4.1. Only now, we identify $\tilde{\mathbf{z}}$ with both $\tilde{\mathbf{z}}^{\mathcal{P}}$ and $\tilde{\mathbf{z}}^{\mathcal{F}}$ and write $\tilde{\mathbf{z}}^{\mathcal{A}}$ with $\mathcal{A} = \mathcal{P} \cup \mathcal{F}$. Using Eq. (18), the guidance signal directly follows from $\mathcal{L}(\hat{\mathbf{z}}_0^{\mathcal{A}}, \tilde{\mathbf{z}}^{\mathcal{P} \cup \mathcal{F}})$ with $\boldsymbol{c}^{\mathcal{P}} = C_{\mathcal{Z}}(\tilde{\mathbf{z}}^{\mathcal{P} \cup \mathcal{F}}, \hat{\mathbf{z}}_0^{\mathcal{P} \cup \mathcal{F}})$. The sampling algorithm is similar to the one described in App. E.1.

Several tasks exist within the FBDD setting [62–65]. Examples are scaffold hopping [64], where the core structure of $\mathbf{z}_0^{\mathcal{M}}$ has to be generated, but functional groups that interact with the receptor are

Table 1: Ligand-Based Drug Design. Results taken from Chen et al. [14] are indicated with ($^*$). We highlight the best conditioning approach for the ShapeMol backbone in **bold** and underline the best approach across all methods.

| | Method | only shape | $\text{Sim}_S$ ($\uparrow$) | $\max \text{Sim}_S$ ($\uparrow$) | $\text{Sim}_G$ ($\downarrow$) | $\max \text{Sim}_G$ ($\downarrow$) | **Ratio** ($\uparrow$) | Diversity ($\uparrow$) |
|---|---|---|---|---|---|---|---|---|
| Non-diffusion based | VS$^*$ [14] | ✗ | $0.729 \pm 0.04$ | $0.807 \pm 0.04$ | $0.226 \pm 0.04$ | $0.241 \pm 0.09$ | $3.226$ | $\underline{0.759 \pm 0.02}$ |
| | SQUID$^*$ [55] ($\lambda = 0.3$) | ✗ | $0.717 \pm 0.08$ | $\underline{0.904 \pm 0.07}$ | $0.349 \pm 0.09$ | $0.549 \pm 0.24$ | $2.054$ | $0.687 \pm 0.07$ |
| | SQUID$^*$ [55] ($\lambda = 1.0$) | ✗ | $0.670 \pm 0.07$ | $0.842 \pm 0.06$ | $0.235 \pm 0.05$ | $0.271 \pm 0.09$ | $2.851$ | $0.744 \pm 0.05$ |
| Diffusion-based | ShapeMol [14] | ✓ | $0.677 \pm 0.04$ | $0.797 \pm 0.04$ | $\mathbf{0.239 \pm 0.05}$ | $0.240 \pm 0.07$ | $2.834$ | $\mathbf{0.714 \pm 0.05}$ |
| | ShapeMol+g [14] | ✗ | $0.744 \pm 0.03$ | $0.849 \pm 0.03$ | $0.242 \pm 0.04$ | $0.245 \pm 0.05$ | $3.074$ | $0.708 \pm 0.05$ |
| | **UniGuide (ShapeMol [U])** | ✓ | $0.726 \pm 0.04$ | $0.827 \pm 0.05$ | $0.248 \pm 0.05$ | $0.239 \pm 0.05$ | $2.927$ | $0.651 \pm 0.05$ |
| | **UniGuide (ShapeMol)** | ✓ | $\mathbf{0.760 \pm 0.05}$ | $\mathbf{0.857 \pm 0.06}$ | $0.240 \pm 0.04$ | $\mathbf{0.237 \pm 0.06}$ | $\mathbf{3.167}$ | $0.705 \pm 0.04$ |
| | **UniGuide (EDM)** | ✓ | $0.749 \pm 0.04$ | $0.860 \pm 0.04$ | $\underline{0.212 \pm 0.04}$ | $\underline{0.206 \pm 0.06}$ | $\underline{3.536}$ | $0.736 \pm 0.04$ |

fixed, or linker design [65], where the connection between separated fragments has to be optimised through the generative process, see Fig. 5. Note that these tasks differ primarily in their application and can be treated identically from a technical perspective within UniGuide. In addition, one can also consider variations where the protein information $\tilde{\mathbf{z}}^{\mathcal{P}}$ is discarded. This usually aligns with switching to an unconditional model $\epsilon_\theta$ that solely models the distribution over molecules. We present results for this configuration in Sec. 5.3.

Furthermore, we would like to highlight that it is possible to combine guidance strategies within UniGuide. For example, one could incorporate a version of the surface condition map $C_{\partial V}$ for FBDD to provide an additional geometric guidance signal for the atoms not included in $\mathcal{F}$.

**Limitations**    Drug discovery also involves tasks beyond purely geometric conditions, encompassing global graph properties [24]. These are excluded from the UniGuide framework. Additionally, UniGuide requires the unconditional model to be trained on a matching configuration space. We discuss the broader impact of our work in App. A.

## 5    Results

In this section, we compare UniGuide to state-of-the-art models across various drug discovery tasks. To highlight the wide range of tasks to which unconditional models can be adapted through UniGuide, we conduct experiments on ligand-based (Sec. 5.1), structure-based (Sec. 5.2) and fragment-based (Sec. 5.3) drug design. We demonstrate that UniGuide performs competitively or even surpasses specialised baseline models, underscoring its practical relevance and transferability to diverse drug discovery scenarios.

### 5.1    Ligand-based drug design

**Dataset**    Following Chen et al. [14], we employ the MOSES dataset for the ligand-based drug design task [66]. We evaluate on a test set consisting of 1000 reference ligands, from which the 3D shape conditions are extracted. For every shape condition $\mathcal{M}_{\text{ref}}$, 50 samples are generated. We refer to App. D.1 for further details on the evaluation setup.

**Baselines**    For the LBDD task, we compare UniGuide to ShapeMol, a conditional diffusion model that is trained by conditioning on learned latent embeddings of the molecular surfaces [14]. Chen et al. [14] also propose a correction technique that adjusts the atom positions based on their distance to the reference ligand's nodes, which is refered to as ShapeMol+g. Additionally, we include as baselines Virtual Screening (VS) [14], a shape-based virtual screening tool, and SQUID [55], a variational autoencoder that decodes molecules by sequentially attaching fragments with fixed bond lengths and angles. For this task, we evaluate UniGuide equipped with the surface condition map $C_{\partial V}$ from Eq. (21) in conjunction with two *unconditionally trained* diffusion models, ShapeMol [U] and EDM [14, 20] as well as the *conditional* model ShapeMol [14]. The "only shape" column in Tab. 1 indicates whether a method uses solely the reference ligand's shape or also incorporates its atom positions.

We compare UniGuide with an alternative guidance approach adapted from Guan et al. [67] in App. D.4 and refer to App. C and App. D.3 for further information on the unconditional models and the guidance parameters, respectively. In addition, inspired by the performance of UniGuide on the

Table 2: Structure-Based Drug Design. Quantitative comparison of generated ligands for target pockets from the CrossDocked and Binding MOAD test sets. Results taken from the respective works are indicated with (*). We highlight the best conditioning approach for the DiffSBDD backbone in **bold** and underline the best approach over all methods.

| | | Method | Vina Score ($\downarrow$) | Vina Min ($\downarrow$) | Vina Dock ($\downarrow$) | QED ($\uparrow$) | SA ($\uparrow$) |
|---|---|---|---|---|---|---|---|
| CrossDocked | Non-Diff. | Test Set | $-6.362 \pm 3.14$ | $-6.707 \pm 2.50$ | $-7.450 \pm 2.33$ | 0.48 | 0.73 |
| | | 3D-SBDD* [38] | $\underline{-5.754 \pm 3.25}$ | $-6.180 \pm 2.42$ | $-6.746 \pm 4.02$ | 0.51 | 0.63 |
| | | Pocket2Mol* [48] | $-5.139 \pm 3.17$ | $-6.415 \pm 2.93$ | $-7.152 \pm 4.90$ | 0.56 | $\underline{0.74}$ |
| | Diffusion-based | DecompDiff* (No Drift) [67] | $-4.750 \pm \ -$ | $-6.170 \pm \ -$ | $-$ | $-$ | $-$ |
| | | TargetDiff* [50] | $-5.466 \pm 8.32$ | $\underline{-6.643 \pm 4.94}$ | $-7.802 \pm 3.62$ | 0.48 | 0.58 |
| | | DiffSBDD-cond [11] | $-3.684 \pm 11.3$ | $-4.670 \pm 6.06$ | $-6.941 \pm 4.33$ | 0.47 | 0.58 |
| | | DiffSBDD [11] | $-4.097 \pm 11.3$ | $-6.306 \pm 5.00$ | $-7.889 \pm 2.61$ | $\underline{\mathbf{0.57}}$ | $\mathbf{0.64}$ |
| | | **UniGuide** | $\mathbf{-5.103 \pm 8.39}$ | $\mathbf{-6.610 \pm 4.20}$ | $\underline{\mathbf{-7.921 \pm 2.43}}$ | $\underline{\mathbf{0.57}}$ | $\mathbf{0.64}$ |
| Binding MOAD | Diffusion-based | Test Set | $-6.748 \pm 2.77$ | $-7.563 \pm 2.53$ | $-8.297 \pm 2.03$ | 0.60 | 0.64 |
| | | DiffSBDD-cond [11] | $-4.466 \pm 2.63$ | $-6.309 \pm 2.52$ | $-7.482 \pm 1.84$ | 0.43 | 0.56 |
| | | DiffSBDD [11] | $-4.744 \pm 7.70$ | $-6.586 \pm 2.59$ | $-7.767 \pm 2.06$ | 0.55 | $\underline{\mathbf{0.62}}$ |
| | | **UniGuide** | $\underline{\mathbf{-5.074 \pm 6.75}}$ | $\underline{\mathbf{-6.622 \pm 2.57}}$ | $\underline{\mathbf{-7.911 \pm 1.97}}$ | $\underline{\mathbf{0.56}}$ | 0.61 |

LBDD task, we further motivate its applicability for the generation of molecules given atom densities, see App. G.

**Evaluation** The goal of LBDD is to discover novel molecules that fit within a given 3D shape. This can be quantified by a high 3D shape similarity and low graph similarity compared to the reference ligand, as illustrated in Fig. 3 as well as App. D.2. We highlight this trade-off by reporting the ratio of these similarities in Tab. 1 as $\text{Sim}_S/\text{Sim}_G$, which constitutes the most important metric for this task. We follow Chen et al. [14] and further evaluate the mean and maximum shape similarities $\text{Sim}_S$ and $\max \text{Sim}_S$, respectively, per reference ligand, measured via the volume overlap between the two aligned molecules. Additionally, we report the graph similarity $\text{Sim}_G$ defined as the Tanimoto similarity between the generated and reference ligand, and the graph similarity $\max \text{Sim}_G$

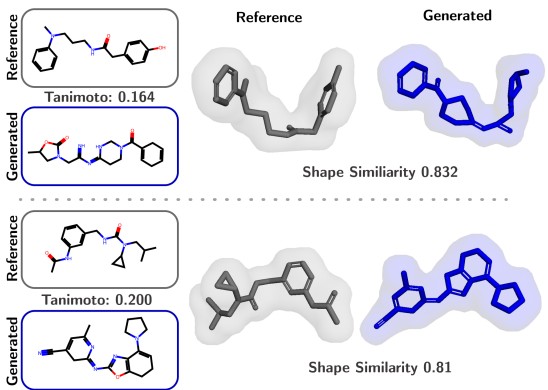

Figure 3: Examples of the two shape-conditioned ligands generated by UniGuide. The goal is to have *low* molecular graph similarity and *high* shape similarity.

of the generated molecule with the maximum shape similarity. Further metrics concerning the quality of the generated ligands are provided in App. D.2.

Both in terms of shape similarity and graph similarity, guiding the generation of EDM with UniGuide outperforms other task-specific conditioning mechanisms and even the Virtual Screening baseline. Emphasised by the Ratio metric across all evaluated methods, UniGuide demonstrates that it is able to generate diverse molecules with very similar shapes compared to the reference ligand. Remarkably, UniGuide achieves higher shape similarity than ShapeMol+g, even though the conditional model is explicitly guided towards the position of the reference ligand through the position correction technique. UniGuide, on the other hand, does not require information about the reference's atom positions at all to generate novel, high-quality ligands. This highlights how UniGuide and the design of condition maps enables unconditional models like EDM, that have not been tailored or trained for the LBDD task, to achieve state-of-the-art performance on new tasks.

## 5.2 Structure-based drug design

**Datasets** Following Schneuing et al. [11], we evaluate UniGuide on two protein-ligand datasets: the CrossDocked dataset [68] and the Binding MOAD dataset [69]. For the CrossDocked dataset, we follow the preprocessing as described by [38] and conduct the evaluation on 100 test protein pockets.

Table 3: Linker Design. Results taken from Igashov et al. [13] are indicated with (*). We underline the best method overall.

| | Method | QED (↑) | SA (↓) | No. Rings (↑) | Valid (↑) | Unique (↑) | 2D Filters (↑) | Recovery (↑) |
|---|---|---|---|---|---|---|---|---|
| Non-diffusion based | DeLinker + ConfVAE + MMFF [53]* | 0.64 ± 0.16 | 3.11 ± 0.68 | 0.21 ± 0.42 | 98.3 | 44.2 | 84.8 | 80.2 |
| | 3DLinker [52]* | 0.65 ± 0.16 | 3.14 ± 0.68 | 0.24 ± 0.43 | 71.5 | 29.2 | 83.7 | 93.5 |
| | 3DLinker (given anchors) [52]* | 0.65 ± 0.16 | 3.11 ± 0.67 | 0.23 ± 0.42 | 99.3 | 29.0 | 84.2 | 94.0 |
| Diffusion-based | DiffLinker [13]* | 0.65 ± 0.15 | 3.19 ± 0.77 | 0.32 ± 0.54 | 90.6 | 51.4 | 87.9 | 70.7 |
| | DiffLinker (given anchors) [13]* | 0.65 ± 0.15 | 3.24 ± 0.81 | 0.36 ± 0.59 | 94.8 | 50.9 | 84.7 | 77.5 |
| | UniGuide (EDM) | 0.64 ± 0.16 | 3.63 ± 1.08 | 0.49 ± 0.62 | 89.1 | 72.1 | 87.9 | 58.8 |

The Binding MOAD dataset is preprocessed as discussed in Schneuing et al. [11], resulting in 130 test proteins. Per target pocket, 100 ligands are generated. We evaluate the generation of ligands on models that are trained on the full-atom context of the pockets in Tab. 2 and results of models trained on the $C_\alpha$ representation of the pockets are provided in App. E.5.

**Baselines**   We compare UniGuide to two autoregressive models designed for the SBDD task: 3D-SBDD [38] and Pocket2Mol [48]. We further include TargetDiff [50] and DecompDiff [67], conditional diffusion models for SBDD that fix the protein pocket context during every step of the diffusion process. We exclude approaches with explicit drift terms like Guan et al. [67] and Huang et al. [70] from the comparison, as UniGuide's SBDD condition map does not include drift terms currently, but can be readily extended to do so. Schneuing et al. [11] present two techniques for controlled structure-based generation: (i) DiffSBDD-cond, a conditional diffusion model similar to [50] and (ii) DiffSBDD, an inpainting-inspired technique that modifies the generative process of an unconditional diffusion model that jointly generates protein-ligand pairs. Across datasets, both UniGuide and DiffSBDD control the same unconditional ligand-protein diffusion model. We provide more information and further evaluation regarding this base model in App. E.2 and App. E.3 and investigate the influence of the guidance scale $S$ as well as the resampling trick [71], a technique that modifies the generative process to better harmonise the generated ligand with the controlled pockets, in App. E.4 and App. E.5.

**Evaluation**   As the task of SBDD is to generate ligands that bind well to a given protein pocket, we assess generated ligands based on affinity-related metrics (Vina Score, Vina Min and Vina Dock), which estimate the binding affinity between the generated ligands and a given test receptor [72]. Additionally, we measure the quality of the generated ligands using two chemical properties: the drug-likeness (QED) and the synthetic accessibility (SA) [66, 73].

Tab. 2 demonstrates that, without additional training or external networks, UniGuide performs competitively with even the highly specialised conditional models like TargetDiff and DecompDiff. Our results indicate that not fully converging to the target protein pocket due to soft guidance, compared to, for example, DiffSBDD's inpainting-inspired technique, is not a limitation in practice. Rather, it suggests that utilising self-guidance in combination with a suitable condition map generates well-harmonised ligand-protein pairs. This is also reflected in the properties of the generated ligands, where UniGuide achieves good drug-likeness (QED) and synthetic accessibility (SA) scores. We provide additional qualitative examples for the SBDD task in Fig. 4, which showcase that UniGuide not only generates drug-like ligands but is even able to improve over the VINA Dock metric of the reference ligand.

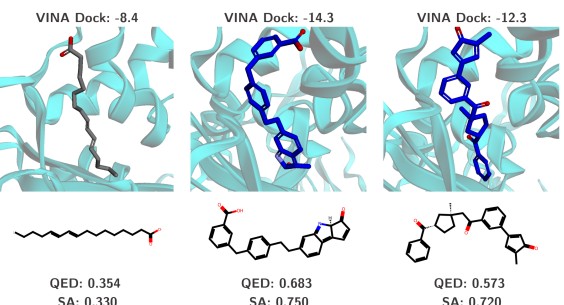

Figure 4: Qualitative example of a test protein pocket (6c0b) from the Binding MOAD dataset. We show the reference ligand (grey) and samples generated by UniGuide (blue).

### 5.3   Fragment-based drug design

**Datasets & Baselines**   In the following, we investigate linker design, a subfield of fragment-based drug design. We follow Igashov et al. [13] and decompose ligands from the ZINC dataset [74] with the MMPA algorithm [75]. Note that the ZINC dataset does not contain pocket information, and the evaluated approaches operate solely at the molecular level. We compare UniGuide to DiffLinker [13], a diffusion-based conditional model that fixes fragments in space. Additionally, we evaluate

the variational autoencoder-based methods DeLinker [53] and 3DLinker [52], adapted as described in Igashov et al. [13]. We provide more information on the experimental setup as well as the unconditionally trained EDM model in App. F.1 and App. C.

**Evaluation** Following Igashov et al. [13], we evaluate the generated linkers and ligands with respect to their properties (SA, QED, Number of Rings and 2D Filters). We additionally measure (i) the uniqueness of the generated samples, (ii) the recovery of the reference ligands, and (iii) the validity, which combines the chemical validity and the successful linking of the fragments.

Using UniGuide to control the EDM generation enables the successful combination of the condition fragments and the generation of diverse linkers. Even compared to task-specific models, UniGuide is able to perform competitively across different metrics. Importantly, UniGuide enables the same unconditional model (EDM) to tackle both the linker design task as presented in Tab. 3 as well as the LBDD task as presented in Tab. 1 without additional training. Note that, while DiffLinker is specifically designed to generate linkers, UniGuide readily generalises to other tasks within the FBDD setting, such as

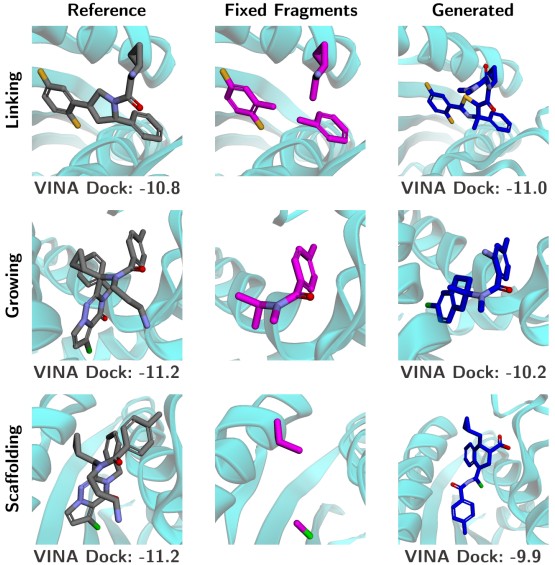

Figure 5: For various pocket-conditioned FBDD tasks, we show reference ligands (grey), desired fragments (magenta), and ligands generated by UniGuide (blue).

fragment growing and scaffolding, see Fig. 5. Additionally, UniGuide is agnostic to the fragmentation procedure used to obtain the condition scaffolds, meaning that UniGuide will generalise to unseen fragments as long as the underlying molecule fits within the training distribution. In App. F.2, we demonstrate how the same unconditional model can be adapted for these tasks. Our quantitative evaluation highlights the benefits achieved through the unification of controlled generation provided by UniGuide.

## 6   Conclusion

In this work, we present UniGuide, a unified way of controlling the generation of molecular diffusion models towards geometric constraints. UniGuide generalises to a multitude of drug discovery tasks without the need for conditioning networks or specialised training protocols, enabling UniGuide to find applicability also in scenarios where little data is available. By demonstrating that specialisation is not a necessity and that a more flexible, unified method outperforms specialised approaches across tasks and datasets, we open up new avenues for streamlined and flexible generative models with wide-ranging applications.

**Acknowledgements** SA, LH, and JS are thankful for valuable feedback from Marcel Kollovieh, Leo Schwinn, and Alessandro Palma from the DAML group and Theis Lab. SA is supported by the DAAD programme Konrad Zuse Schools of Excellence in Artificial Intelligence, sponsored by the Federal Ministry of Education and Research. LH is supported by the Helmholtz Association under the joint research school "Munich School for Data Science - MUDS". FJT acknowledges support from the Helmholtz Association's Initiative and Networking Fund through Helmholtz AI (ZT-I-PF-5-01). FJT further acknowledges support by the BMBF (01IS18053A). In addition, FJT consults for Immunai Inc., Singularity Bio B.V., CytoReason Ltd, and Omniscope Ltd and has an ownership interest in Dermagnostix GmbH and Cellarity.

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

## A Impact Statement

Our research holds the promise of significant contributions to the advancement of drug discovery, possibly assisting in the discovery of novel pharmaceutical compounds. Nevertheless, because of its applications in drug discovery, this strategy is not without its hazards. The ability to produce various molecules with desired properties may not only serve the purpose of beneficial drug development but may also unintentionally result in the creation of dangerous substances or compounds with unexpected effects. These concerns underline the critical need for careful handling when working with the structures this method can generate.

## B Proof of Theorem 4.1

First, recall Theorem 4.1 that we provide in Sec. 4:

**Theorem 1.** *Consider a function $C : \mathcal{S} \times \mathcal{Z} \to \mathcal{Z}$. If $C(\boldsymbol{s}, \mathbf{z})$ is invariant to rigid transformations $G$ in the first argument and equivariant in the second argument, then the gradient $\nabla_{\mathbf{z}} \|\boldsymbol{v}\|_2^2$ of the vector $\boldsymbol{v} = \mathbf{z} - C(\boldsymbol{s}, \mathbf{z})$ is equivariant to transformations of $\mathbf{z}$.*

*Proof.* We start the proof by showing that $\|\boldsymbol{v}\|_2$ is invariant to transformations of both $\mathbf{z}$ and $\boldsymbol{s}$.

1. $\|\mathbf{z} - C(\boldsymbol{s}, \mathbf{z})\|_2$ is invariant to transformations in $\boldsymbol{z}$:

$$
\begin{aligned}
\left\|G\mathbf{z} - C(\boldsymbol{s}, G\mathbf{z})\right\|_2 &= \left\|G\mathbf{z} - GC(\boldsymbol{s}, \mathbf{z})\right\|_2 && [C \text{ is equivariant in } \boldsymbol{z}] \\
&= \left\|G(\mathbf{z} - C(\boldsymbol{s}, \mathbf{z}))\right\|_2 && (22) \\
&= \left\|\mathbf{z} - C(\boldsymbol{s}, \mathbf{z})\right\|_2 && [G \text{ is a rigid transformation}]
\end{aligned}
$$

2. $\|\mathbf{z} - C(\boldsymbol{s}, \mathbf{z})\|_2$ is invariant to transformations $\boldsymbol{s}$ follows immediately:

$$
\left\|\mathbf{z} - C(G\boldsymbol{s}, \mathbf{z})\right\|_2 = \left\|\mathbf{z} - C(\boldsymbol{s}, \mathbf{z})\right\|_2 \qquad [C \text{ is invariant in } \boldsymbol{s}] \tag{23}
$$

In a second step, we make use of the fact that for a group of transformations $G$, it holds that if $\mathcal{L}(\cdot, \cdot)$ is a $G$-invariant function, $\nabla_x \mathcal{L}(\cdot, x)$ is $G$-equivariant [76]. From the invariance of $\|\boldsymbol{v}\|_2$, it follows immediately that $\nabla_{\mathbf{z}} \|\mathbf{z} - C(\boldsymbol{s}, \mathbf{z})\|_2^2$ is equivariant to transformations of $\mathbf{z}$. □

## C Unconditional Equivariant Diffusion Model

UniGuide guides an unconditional diffusion model given an arbitrary condition and a natural choice for a model operating only on the molecule level is the EDM model as proposed in Hoogeboom et al. [20].

We adapt this model for two tasks presented in this work, namely the LBDD task discussed in Sec. 5.1 and the Linker Design task as presented in Sec. 5.3. For these tasks, we train an unconditional EDM model both on the MOSES dataset [66] in the configuration as described in Chen et al. [14] and on the ZINC dataset [74] as described in Igashov et al. [13]. For both trainings, we employ the hyperparameter configuration for the GEOM dataset as described in Hoogeboom et al. [20]. We run multi-GPU trainings on 4 NVIDIA A100 GPUs until convergence, however, a single NVIDIA A100 GPU is sufficient for this training and will only increase the training time. For inference, we employ the Resampling trick as discussed in Lugmayr et al. [71] with $R = 10$ resampling steps and $T = 100$ timesteps. EDM is available under the MIT License.

## D Ligand-based drug design

### D.1 Implementation details

We train two unconditional diffusion models, ShapeMol [U] and EDM, to generate 3D molecules on the MOSES dataset [66], licensed under the MIT License, for which we generate 3D conformers with RDKit [73], available under the BSD 3-Clause License. We use $1,593,653$ training samples

and randomly select 1000 samples for validation. The model architecture of ShapeMol[U] is an unconditional version of the ShapeMol model proposed in Chen et al. [14], and it is trained with 1000 diffusion steps. ShapeMol [U] is trained with a batch size of 32 on two NVIDIA A100 GPUs for 500 epochs. Unlike ShapeMol, we do not concatenate the molecular surface embedding of the ligands to the features. For the shape-conditioned generation with position correction (ShapeMol+g), we follow the scheme proposed in Chen et al. [14]. It provides further guidance to the conditional generation by sampling 20 query points from a Gaussian distribution centred around every atom in the reference ligand. The position correction adjusts the coordinates of the predicted atom positions during every generation step by pushing the coordinates close to the query points as follows:

$$\hat{\boldsymbol{x}} = (1 - \sigma)\hat{\boldsymbol{x}} + \sigma \sum_{\mathbf{z} \in n(\hat{\boldsymbol{x}}, \mathcal{Q})} \mathbf{z}/n, \text{if} \sum_{\mathbf{z} \in n(\hat{\boldsymbol{x}}, \mathcal{Q})} d(\hat{\boldsymbol{x}}, \mathbf{z})/n > \gamma, \quad (24)$$

where $d(\hat{\boldsymbol{x}}, \mathbf{z})$ is the Euclidean distance, $n(\hat{\boldsymbol{x}}, \mathcal{Q})$ is the set of $n$ nearest neighbors of $\hat{\boldsymbol{x}}$ in $\mathcal{Q}$ and $\gamma > 0$ is a distance threshold. We follow the implementation of Chen et al. [2] for the position correction method by setting $\gamma = 0.2$ and only guiding during the first 700 denoising steps.

For the shape-conditioned generation with UniGuide, we extract the mesh of the condition ligand using the Open Drug Discovery Toolkit [77], which is available under the BSD 3-Clause revised License. The query points we use for guidance are 512 points sampled uniformly on the mesh surface. For the evaluation, we measure the shape similarity $\text{Sim}_S$ as the volume overlap between the aligned generated ligand and the condition ligand. For the alignment, we utilise the ShaEP tool [78].

We provide a detailed description of the LBDD sampling algorithm in Algorithm 1.

---

Algorithm 1: Sampling algorithm to generate a ligand that is conditioned on a reference ligand $\mathcal{M}_{\text{ref}}$'s surface, using an unconditional model $\boldsymbol{\epsilon}_\theta(\mathbf{z}_t, t)$ modelling the distribution over molecules. The points $\mathbf{y} \in \mathbb{R}^{K \times 3}$ are sampled uniformly from the surface of $\mathcal{M}_{\text{ref}}$, enclosing the volume $V$.

---

**Require:** $\mathbf{y}, \alpha$: desired margin to surface, $k$: number of nearest neighbours
$\mathbf{z}_T \sim \mathcal{N}(\mathbf{0}, \boldsymbol{I})$                                   {Sample from normal prior}
**for** $t = T$ **to** 1 **do**
    $\mathbf{x}_t, \mathbf{h}_t = \mathbf{z}_t$
    $\hat{\mathbf{x}}_0 = \frac{\mathbf{x}_t - \sqrt{1 - \bar{\alpha}_t} \boldsymbol{\epsilon}_\theta^{\mathbf{x}}(\mathbf{z}_t, t)}{\sqrt{\bar{\alpha}_t}}$     {Compute the conformation $\hat{\mathbf{x}}_0$ of the clean approximation $\hat{\mathbf{z}}_0$}
    For every atom $\hat{\boldsymbol{x}}_i$ in $\hat{\mathbf{x}}_0$ do:
        $\bar{\boldsymbol{y}}_i = \frac{1}{k} \sum_{\boldsymbol{y} \in \mathcal{N}_{\hat{\boldsymbol{x}}_i}} \boldsymbol{y}$            {Compute the mean of $k$ nearest neighbors of $\hat{\boldsymbol{x}}_i$ in $\mathbf{y}$}
        Compute $(\boldsymbol{c_{\mathbf{x}}})_i$ based on Eq. (21)            {Compute component-wise condition map}
    $\mathcal{L} = \mathcal{L}(\hat{\mathbf{x}}_0, \boldsymbol{c_{\mathbf{x}}})$
    $\boldsymbol{g} = \nabla_{\mathbf{x}_t} \mathcal{L}$                                        {Compute gradient of guidance loss}
    $\boldsymbol{\mu}_t = \boldsymbol{\mu}_\theta(\mathbf{z}_t, t) - \lambda(t) \cdot \boldsymbol{g}$                         {Update the mean function}
    $\mathbf{z}_{t-1} \sim \mathcal{N}(\boldsymbol{\mu}_t, \sigma_t \boldsymbol{I})$
**end for**
**return** $\mathbf{z}_0$

---

## D.2 Additional results

For completeness, we report additional quantitative evaluation of the generated ligands' properties in Tab. 4. We also provide further qualitative results of the generated ligands for the LBDD task in Fig. 6. UniGuide generates ligands with better shape similarity to the reference ligands compared to the conditional model ShapeMol with the position correction technique.

Table 4: Additional ligand property results for the methods discussed in Sec. 5.1. We report mean and standard deviation and highlight the best result in **bold**.

| method | Connect. (↑) | Unique (↑) | QED | SA (↑) | LogP (↑) | Lipinski (↑) |
|---|---|---|---|---|---|---|
| ShapeMol | 98.8% | 99.9% | **0.753** | $0.640 \pm 0.104$ | $2.001 \pm 1.360$ | $4.979 \pm 0.156$ |
| ShapeMol+g | 97.0% | 99.8% | 0.751 | $0.630 \pm 0.110$ | $1.908 \pm 1.508$ | $4.874 \pm 0.170$ |
| **UniGuide+ ShapeMol[U]** | 98.0% | **100%** | 0.736 | $0.625 \pm 0.103$ | $1.828 \pm 1.463$ | $4.974 \pm 0.186$ |
| **UniGuide (ShapeMol)** | 99.0% | **100%** | 0.750 | $\mathbf{0.641 \pm 0.107}$ | $\mathbf{2.002 \pm 1.374}$ | $4.982 \pm 0.152$ |
| **UniGuide+ EDM** | **99.8%** | 99.99% | 0.742 | $0.636 \pm 0.088$ | $1.833 \pm 1.221$ | $\mathbf{4.994 \pm 0.082}$ |

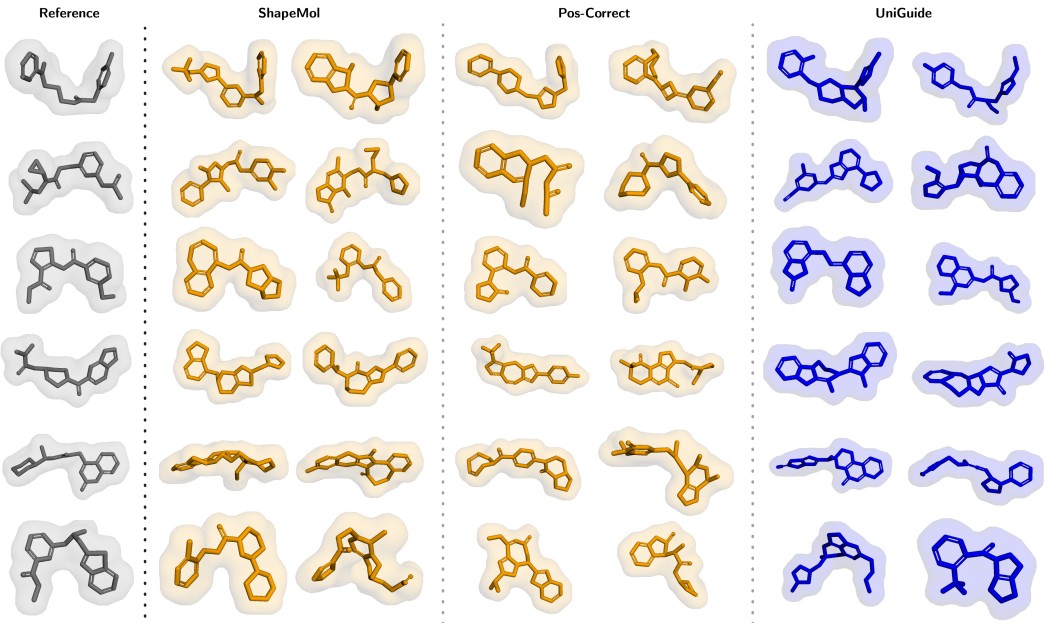

Figure 6: Examples of the ligands generated by *ShapeMol*, Pos-Correct and UniGuide. Pos-Correct is the position correction technique proposed by Chen et al. [14]. Both Pos-Correct and UniGuide are combined with the unconditionally trained model ShapeMol [U]. We plot the reference ligand as well as the generated ligands with their shapes.

Table 5: Comparison of UniGuide with validity guidance for shape-based generation. We highlight the ratio metric as the most critical indicator, reflecting the balance between shape similarity and graph dissimilarity.

| | $\text{Sim}_S$ (↑) | $\text{maxSim}_S$ (↑) | $\text{Sim}_G$ (↓) | $\text{maxSim}_G$ (↓) | **Ratio** (↑) | Connect. (↑) | Unique. (↑) | Diversity (↑) | QED (↑) |
|---|---|---|---|---|---|---|---|---|---|
| Validity Guidance | 0.59 | 0.76 | 0.20 | 0.20 | **2.96** | 97% | 100% | 0.76 | 0.69 |
| UniGuide (EDM) | 0.74 | 0.86 | 0.21 | 0.20 | **3.53** | 99% | 99% | 0.73 | 0.74 |

### D.3 Guidance parameters

For the LBDD task, the guidance strength $S$ is weighted by an exponentially decreasing function $\frac{\beta_t}{\sqrt{\alpha_t}}$. For the guided generation using the unconditional ShapeMol [U] model under the UniGuide framework, we define a scale scheduler that increases with an exponent of $1.01$ and weight it with $\frac{\beta_t}{\sqrt{\alpha_t}}$ and guide from the diffusion step 1000 to the diffusion step 200. For the guided generation using the EDM model, we use a linear scale function that increases from 5 to 15. The guidance is applied from the diffusion step 920 to the last timestep 1.

### D.4 Comparison of UniGuide with an alternative loss formulation

We adapt the validity guidance loss from Guan et al. [50] to the LBDD setting. The proposed loss is grounded in the smooth distance function $S(x)$ from Sverrisson et al. [79], which computes as:

$$S(x) = -\sigma \log \Big( \sum_{i}^{N} \exp(-\|x_t - y_i\|_2^2 / \sigma) \Big) \quad .$$

This function provides an alternative approach to shape-based generation by deriving an appropriate loss function $\sum_x S(x)$, rather than modifying the condition map as proposed by UniGuide. Here, $S(x)$ implicitly defines a surface through $S(x) - \gamma = 0$ and points $x_t$ inside satisfy $S(x_t) < \gamma$.

On a technical level, the gradient for validity guidance computes as follows:

$$\nabla_{x_t} S(x_t) = \nabla_{x_t} \Big[ -\sigma \log \Big( \sum_i^N \exp(-\|x_t - y_i\|_2^2/\sigma) \Big) \Big]$$

$$= \frac{1}{\sum \exp(\ldots)} \sum_i^N \underbrace{\exp(\ldots)}_{\omega_i} \nabla_{x_t} \|x_t - y_i\|_2^2$$

$$= \frac{1}{\sum \omega_i} \nabla_{x_t} \sum_i^N \omega_i \|x_t - y_i\|_2^2 \quad .$$

This gradient formulation is quite similar (up to the weighting) to UniGuide's special case $\mathcal{S} = \mathcal{Z}$, as it computes an $L_2$ loss on a given conformation ($\{y_i\}$), see Eq. (18), meaning that it does not generalise to arbitrary geometric conditions.

We emphasise that UniGuide is more broadly applicable because it separates surface computation from gradient computation, offering two key benefits. First, since the condition map does not require differentiability, there is greater flexibility in computing surface points. Second, the precise geometric intuition behind the condition map makes it easier to adapt to new scenarios, as demonstrated by our application to generating density-guided molecules.

For the empirical comparison, we selected the hyperparameters $\sigma$ and $\gamma$ in the surface loss computation to achieve a high DICE score between the implicitly defined surface and the meshes UniGuide utilises for LBDD ($\sigma = 1$, $\gamma = 2$, DICE $> 0.8$). Our surface calculations use the Open Drug Discovery Toolkit (ODDT), which assigns specific radii to individual atom types and employs the marching cubes algorithm to generate meshes [80].

We performed several runs around the above-specified hyperparameter configuration. The runs performed similarly, and we report the best result in Tab. 5. Although validity guidance for LBDD yields low graph similarity, the shape similarity remains suboptimal compared to UniGuide. Additionally, we frequently encounter numerical instability when computing the guidance term, an issue not present with UniGuide's formulation of LBDD. One possible explanation for this numerical instability is that the surface is defined implicitly, unlike UniGuide where it is explicitly defined. The explicit definition in UniGuide allows for relating the gradient updates directly to the surface, as shown in Eq. (21).

## E   Structure-based drug design

---

Algorithm 2: Sampling algorithm to generate a ligand conditioned on a protein pocket $\tilde{\mathbf{z}}^{\mathcal{P}}$ using the unconditional joint model $\epsilon_\theta(\mathbf{z}_t, t)$, where $\mathbf{z}_t = [\mathbf{z}_t^{\mathcal{M}}, \mathbf{z}_t^{\mathcal{P}}]$, that models the distribution $P(\mathbf{z}^{\mathcal{M}}, \mathbf{z}^{\mathcal{P}})$. The guidance signal is controlled via the guidance strength $S$. Note that samples from the generative process $p_\theta(\mathbf{z}_{t-1}|\mathbf{z}_t)$ are assumed to be CoM-free.

---

**Require:** $\tilde{\mathbf{z}}^{\mathcal{P}}$, $S$
$\mathbf{z}_T \sim \mathcal{N}(\mathbf{0}, \boldsymbol{I})$ {Sample from normal prior}
**for** $t = T$ **to** 1 **do**
$\quad \hat{\mathbf{z}}_0^{\mathcal{P}} = \mathbf{z}_t^{\mathcal{P}} - \sqrt{1 - \bar{\alpha}_t} \epsilon_\theta^{\mathcal{P}}(\mathbf{z}_t, t)/\sqrt{\bar{\alpha}_t}$ {Compute the clean data of the pocket}
$\quad \mathcal{L} = \mathcal{L}(\hat{\mathbf{z}}_0^{\mathcal{P}}, \tilde{\mathbf{z}}^{\mathcal{P}})$
$\quad \boldsymbol{g} = (\nabla_{\mathbf{x}_t} \mathcal{L} - \overline{\nabla_{\mathbf{x}_t} \mathcal{L}}, \nabla_{\mathbf{h}_t} \mathcal{L})$ {Compute gradient and substract the CoM}
$\quad \boldsymbol{\mu}_t = \boldsymbol{\mu}_\theta(\mathbf{z}_t, t) - \lambda(t) \cdot \boldsymbol{g}$ {Update the mean of the pocket}
$\quad \mathbf{z}_{t-1} \sim \mathcal{N}(\boldsymbol{\mu}_t, \sigma_t \boldsymbol{I})$
**end for**
**return** $\mathbf{z} = (\mathbf{z}_0^{\mathcal{M}}, \mathbf{z}_0^{\mathcal{P}})$

---

### E.1   SBDD sampling algorithm

We provide the algorithm for inference in the SBDD task scenario in Algorithm 2.

## E.2 Ligand-protein generative joint model

SBDD aims to generate a ligand given a protein pocket: $p_\theta(\mathbf{z}^\mathcal{M} \mid \mathbf{z}^\mathcal{P}_{\text{test}}, t)$. We adopt DiffSBDD [11], an unconditional joint diffusion model that approximates the joint distribution $p(\mathbf{z}^\mathcal{M}_{\text{data}}, \mathbf{z}^\mathcal{P}_{\text{data}})$ of generating ligand-protein pairs, where the noise predictor $\boldsymbol{\epsilon}_\theta(\mathbf{z}^\mathcal{M}_t, \mathbf{z}^\mathcal{P}_t, t)$ is parametrised by EGNN. DiffSBDD is available under the MIT License. To process ligand and pocket nodes with a single GNN, atom types and residue types are embedded jointly. Atom and residue features are then decoded separately using atom decoder and residue decoder to $\boldsymbol{\epsilon}^\mathcal{M}_\theta(\mathbf{z}^\mathcal{M}_t, \mathbf{z}^\mathcal{P}_t, t)$ and $\boldsymbol{\epsilon}^\mathcal{P}_\theta(\mathbf{z}^\mathcal{M}_t, \mathbf{z}^\mathcal{P}_t, t)$ [11].

For the unconditional sampling with the joint model, the number of ligand and pocket nodes is sampled from the joint node distribution $p(N^\mathcal{M}, N^\mathcal{P})$, measured across a training set of $(\mathcal{M}, \mathcal{P})$ pairs. During the modified generative process with the inpainting-inspired technique or with UniGuide the number of pocket nodes is set to be equal to the number of nodes in $\mathcal{P}_{\text{test}}$, while the size of the ligand is generated from a conditional distribution $p(N^\mathcal{M} \mid N^\mathcal{P})$. Since this sampling procedure leads to ligands that are much smaller compared to the reference ligands found in the test set, the mean size of sampled ligands is increased by 10 for Binding MOAD and 5 for CrossDocked during ligand generation [11]. We utilize the unconditional base models from Schneuing et al. [11], which are trained on either the $C_\alpha$ or full-atom context from the Binding MOAD or CrossDocked datasets. However, we retrain the DiffSBDD model specifically on the full-atom context of the CrossDocked data, as we were unable to reproduce the reported results in this configuration from Schneuing et al. [11]. We find that contrary to what is reported in Schneuing et al. [11], the model converges early and does not need a full 1000 epochs to fully train. We employ this checkpoint to evaluate both the DiffSBDD inpainting-inspired approach as well as UniGuide. We train the model on four NVIDIA A100 GPU with a batch size of 2. 8 training epochs take approximately 24 hours.

Table 6: Hyperparameters of ligand and proteins graphs in joint models

|  | CROSSDOCKED | | BINDING MOAD | |
| --- | --- | --- | --- | --- |
|  | JOINT $C_\alpha$ MODEL | JOINT FULL-ATOM MODEL | JOINT $C_\alpha$ MODEL | JOINT FULL-ATOM MODEL |
| EDGES (LIGAND-LIGAND) | FULLY CONNECTED | FULLY CONNECTED | FULLY CONNECTED | FULLY CONNECTED |
| EDGES (LIGAND-POCKET) | $< 5$ Å | $< 5$ Å | $< 8$ Å | $< 7$ Å |
| EDGES (POCKET-POCKET) | $< 5$ Å | $< 5$ Å | $< 8$ Å | $< 4$ Å |

**Representing ligands and proteins as graphs**   Proteins consist of amino acids, where every amino acid is a set of amino $(NH)$, carboxyl $(CO)$, $\alpha$-carbon atom and a side chain $(R)$ that is specific to every amino acid type [81]. The $C_\alpha$-representation of a protein pocket is a residue-level graph, in which the node features of the protein are represented as one-hot encodings of the amino acid type. The full-atom representation of the receptor is an atom-level graph and represents the full context of the protein pocket. Details on processed graphs of the join model $p(\mathbf{z}^\mathcal{M}, \mathbf{z}^\mathcal{P})$ are provided in Tab. 6. We refer the reader to Schneuing et al. [11] for more information on the hyperparameters of the joint model.

Table 7: Quantitative evaluation of samples generated by the unconditional joint models [11] trained on Crossdocked (C.D.) and Binding MOAD (B.M). We report the mean over all generated ligands.

| DATASET | $R$ | $T$ | QED ($\uparrow$) | SA ($\uparrow$) | LIPINSKI ($\uparrow$) | DIVERSITY ($\uparrow$) | CONNECTIVITY ($\uparrow$) | VALIDITY ($\uparrow$) |
| --- | --- | --- | --- | --- | --- | --- | --- | --- |
| C.D. ($C_\alpha$) | 1 | 500 | 0.535 | 0.660 | 4.741 | 0.772 | 0.893 | 0.986 |
| C.D. ($C_\alpha$) | 10 | 50 | 0.578 | 0.752 | 4.836 | 0.774 | 0.994 | 0.986 |
| B.M. ($C_\alpha$) | 1 | 500 | 0.471 | 0.608 | 4.783 | 0.824 | 0.839 | 0.985 |
| B.M. ($C_\alpha$) | 10 | 50 | 0.544 | 0.665 | 4.883 | 0.823 | 0.961 | 0.992 |

## E.3 Further Comparison to DiffSBDD

In addition to Tab. 2, we follow the experimental setup as utilised in Schneuing et al. [11] to compare UniGuideto DiffSBDD, which uses the same base model, in particular. In Tab. 8, we further investigate the advantages of using self-guidance in combinations with UniGuide over both the conditional DiffSBDD model (DiffSBDD-cond) as well as the inpainting-inspired technique

Table 8: Quantitative comparison of generated ligands for target pockets from the CrossDocked and Binding MOAD test sets. Results taken from Schneuing et al. [11] are indicated with ($^*$). We report mean and standard deviation and highlight the best diffusion-based approach in **bold**.

| | | Vina ($\downarrow$) | Vina Top 10% ($\downarrow$) | QED ($\uparrow$) | SA ($\uparrow$) | Lipinski ($\uparrow$) | Diversity ($\uparrow$) | RMSD ($\downarrow$) |
|---|---|---|---|---|---|---|---|---|
| CrossDocked | Test Set | $-6.865 \pm 2.35$ | - | $0.476 \pm 0.20$ | $0.728 \pm 0.14$ | $4.340 \pm 1.14$ | - | - |
| | 3D-SBDD* [38] | $-5.888 \pm 1.91$ | $-7.289 \pm 2.34$ | $0.502 \pm 0.17$ | $0.675 \pm 0.14$ | $4.787 \pm 0.51$ | $0.742 \pm 0.09$ | - |
| | Pocket2Mol* [48] | $-7.058 \pm 2.80$ | $-8.712 \pm 3.18$ | $0.572 \pm 0.16$ | $0.752 \pm 0.12$ | $4.936 \pm 0.27$ | $0.735 \pm 0.15$ | - |
| | Graph-BP* [49] | $-4.719 \pm 4.03$ | $-7.165 \pm 1.40$ | $0.502 \pm 0.12$ | $0.307 \pm 0.09$ | $4.883 \pm 0.37$ | $0.844 \pm 0.01$ | - |
| | TargetDiff* [50] | $-7.318 \pm 2.47$ | $\mathbf{-9.669 \pm 2.55}$ | $0.483 \pm 0.20$ | $0.584 \pm 0.13$ | $4.594 \pm 0.83$ | $0.718 \pm 0.09$ | $\mathbf{0.000 \pm 0.00}$ |
| | DiffSBDD-cond* | $-6.950 \pm 2.06$ | $-9.120 \pm 2.16$ | $0.469 \pm 0.21$ | $0.578 \pm 0.13$ | $4.562 \pm 0.89$ | $\mathbf{0.728 \pm 0.07}$ | $\mathbf{0.000 \pm 0.00}$ |
| | DiffSBDD | $-7.216 \pm 2.54$ | $-9.490 \pm 2.00$ | $\mathbf{0.571 \pm 0.19}$ | $\mathbf{0.639 \pm 0.14}$ | $4.808 \pm 0.50$ | $0.707 \pm 0.09$ | $0.045 \pm 0.01$ |
| | **UniGuide** | $\mathbf{-7.320 \pm 2.27}$ | $-9.514 \pm 2.04$ | $\mathbf{0.571 \pm 0.19}$ | $0.638 \pm 0.14$ | $\mathbf{4.822 \pm 0.47}$ | $0.705 \pm 0.08$ | $0.047 \pm 0.01$ |
| Bind. MOAD | Test Set | $-8.331 \pm 2.05$ | - | $0.602 \pm 0.15$ | $0.636 \pm 0.08$ | $4.838 \pm 0.37$ | - | - |
| | Graph-BP* [49] | $-4.843 \pm 2.24$ | $-6.629 \pm 0.95$ | $0.512 \pm 0.11$ | $0.310 \pm 0.09$ | $4.945 \pm 0.27$ | $0.826 \pm 0.01$ | $\mathbf{0.000 \pm 0.00}$ |
| | DiffSBDD-cond | $-7.172 \pm 1.88$ | $-9.174 \pm 2.13$ | $0.430 \pm 0.20$ | $0.564 \pm 0.12$ | $4.526 \pm 0.80$ | $0.711 \pm 0.08$ | $\mathbf{0.000 \pm 0.00}$ |
| | DiffSBDD | $-7.263 \pm 4.19$ | $-9.776 \pm 2.25$ | $0.546 \pm 0.21$ | $\mathbf{0.618 \pm 0.12}$ | $4.777 \pm 0.54$ | $\mathbf{0.740 \pm 0.05}$ | $53 \pm 31$ |
| | **UniGuide** | $\mathbf{-7.661 \pm 2.99}$ | $\mathbf{-9.864 \pm 2.13}$ | $\mathbf{0.556 \pm 0.20}$ | $0.605 \pm 0.12$ | $\mathbf{4.799 \pm 0.50}$ | $0.723 \pm 0.05$ | $55 \pm 31$ |

(DiffSBDD). UniGuide reliably achieves superior VINA Dock scores compared to both DiffSBDD models and performs competitively with the conditional TargetDiff model. In App. E.4 and App. E.5, we expand on this experimental comparison with further analysis of the effects of Resampling as well as the guidance strength.

## E.4 Resampling

Inpainting is introduced for diffusion models to condition outputs with fixed parts [71] and can be applied for structure-based molecular tasks. Given a model that generates $(\mathbf{z}_t^{\mathcal{M}}, \mathbf{z}_t^{\mathcal{P}})$ pairs at denoising step $t$, the protein pocket $P_t$ is replaced with the noised representation of protein context $\tilde{\mathbf{z}}_t^{\mathcal{P}}$. This noised representation can be obtained through the forward process of diffusion models as specified in Eq. (2). However, the direct application of this method leads to locally harmonised samples that struggle to incorporate the global context [71]. In order to effectively harmonise the generated information during the entire generative process, Lugmayr et al. [71] propose a technique they call "Resampling". This modifies the reverse Markov chain by moving back and forth in the diffusion process to enable the model to better incorporate the replaced components.

Schneuing et al. [11] propose to use the same resampling technique to harmonise the replaced protein context with the ligand, since the replaced receptor is sampled independently of the ligand. During resampling, each latent representation is repeatedly diffused back and forth before advancing to the next time step. We found that resampling further improves the general performance of the unconditional generation, and thus improves the guided generation as well. We report results for this in App. E.5, where we evaluate how the unconditional generation of the joint model is improved across different metrics with added resampling steps. We follow Schneuing et al. [11] in using the setting of $R = 10$ resampling steps and $T = 50$ timesteps. While DiffSBDD resamples the ligand and the noised target protein pocket, we resample the guided protein pocket and ligand with UniGuide. In general, the concept of resampling can be applied to harmonise the configuration $\mathbf{z}_t$ with the condition $\mathbf{c}$.

## E.5 Guidance parameters

The guidance scale $S$ controls the strength of the guiding signal, see Eq. (7) and it is weighted by $w(t) = \frac{\beta(t)}{\sqrt{\alpha_t}}$ during the generation. We use a constant scale $S$ for structure-based drug design experiments and evaluate for several guidance scale values in Tab. 9 and Tab. 10 for models trained on the Binding MOAD dataset with $C_\alpha$ and full-atom representation respectively. The quantitative evaluation on the CrossDocked data is shown under Tab. 11 and Tab. 12 with additional metrics reported in Tab. 7. For the generation with the $C_\alpha$-models, we generate 100 samples for every test pocket with a batch size of 50. The full generation takes approximately 5 hours for Binding MOAD and 6 hours for CrossDocked. For the DiffSBDD model trained on the Binding MOAD fullatom pocket data, we use a batch size of 15 for the generation. We use a batch size of 2 to sample with the DiffSBDD model trained on CrossDocked (fullatom).

Table 9: Results for the Binding MOAD test set with the unconditional DiffSBDD base model trained on the $C_\alpha$-representation of the pockets combined with UniGuide and the inpainting-inspired technique DiffSBDD [11]. We provide results for varying the guidance scales $S$ during our controlled generation. We also report results for the DiffSBDD-cond ($C_\alpha$) model trained on the $C_\alpha$ pockets.

| METHOD | $S$ | $R/T$ | VINA (↓) | VINA TOP 10% (↓) | QED (↑) | SA (↑) | LIPINSKI (↑) | DIVERSITY (↑) | RMSD (↓) |
|---|---|---|---|---|---|---|---|---|---|
| DIFFSBDD-COND ($C_\alpha$) | - | - | -6.628 ± 1.59 | -8.291 ± 1.26 | 0.481 ± 0.20 | 0.554 ± 0.11 | 4.651 ± 0.70 | 0.714 ± 0.04 | 0.000 ± 0.00 |
| DIFFSBDD | - | 1/500 | -6.362 ± 3.04 | -8.179 ± 1.24 | 0.452 ± 0.20 | 0.541 ± 0.11 | 4.604 ± 0.76 | 0.734 ± 0.03 | 0.008 ± 0.01 |
| UNIGUIDE | 1.0 | 1/500 | -6.519 ± 2.05 | -8.227 ± 1.23 | 0.464 ± 0.20 | 0.540 ± 0.11 | 4.627 ± 0.73 | 0.733 ± 0.03 | 0.125 ± 0.01 |
| UNIGUIDE | 2.0 | 1/500 | -6.568 ± 2.13 | -8.268 ± 1.25 | 0.471 ± 0.20 | 0.543 ± 0.11 | 4.636 ± 0.73 | 0.735 ± 0.04 | 0.105 ± 0.25 |
| UNIGUIDE | 3.0 | 1/500 | -6.667 ± 1.92 | -8.305 ± 1.28 | 0.468 ± 0.20 | 0.542 ± 0.11 | 4.622 ± 0.73 | 0.737 ± 0.03 | 0.072 ± 0.03 |
| UNIGUIDE | 4.0 | 1/500 | -6.587 ± 1.86 | -8.293 ± 1.29 | 0.470 ± 0.20 | 0.544 ± 0.11 | 4.636 ± 0.72 | 0.735 ± 0.03 | 0.058 ± 0.01 |
| UNIGUIDE | 6.0 | 1/500 | -6.568 ± 1.93 | -8.284 ± 1.26 | 0.468 ± 0.20 | 0.542 ± 0.11 | 4.630 ± 0.73 | 0.734 ± 0.03 | 0.045 ± 0.01 |
| UNIGUIDE | 7.0 | 1/500 | -6.575 ± 1.86 | -8.296 ± 1.28 | 0.469 ± 0.20 | 0.544 ± 0.11 | 4.636 ± 0.72 | 0.735 ± 0.03 | 0.043 ± 0.05 |
| DIFFSBDD | - | 10/50 | -6.896 ± 3.10 | -8.962 ± 1.37 | 0.547 ± 0.20 | 0.578 ± 0.20 | 4.754 ± 0.50 | 0.709 ± 0.05 | 0.007 ± 0.01 |
| UNIGUIDE | 1.0 | 10/50 | -6.845 ± 3.68 | -8.972 ± 1.36 | 0.547 ± 0.19 | 0.578 ± 0.13 | 4.756 ± 0.53 | 0.709 ± 0.05 | 0.216 ± 0.21 |
| UNIGUIDE | 2.0 | 10/50 | -6.889 ± 3.83 | -9.018 ± 1.40 | 0.547 ± 0.19 | 0.577 ± 0.13 | 4.756 ± 0.52 | 0.707 ± 0.04 | 0.279 ± 0.03 |
| UNIGUIDE | 3.0 | 10/50 | -7.050 ± 2.38 | -9.051 ± 1.39 | 0.551 ± 0.18 | 0.575 ± 0.14 | 4.763 ± 0.50 | 0.706 ± 0.04 | 0.220 ± 0.01 |
| UNIGUIDE | 4.0 | 10/50 | -7.016 ± 2.93 | -9.023 ± 1.38 | 0.552 ± 0.18 | 0.578 ± 0.14 | 4.765 ± 0.50 | 0.708 ± 0.03 | 0.168 ± 0.05 |
| UNIGUIDE | 6.0 | 10/50 | -7.053 ± 2.91 | -9.067 ± 1.39 | 0.550 ± 0.18 | 0.579 ± 0.14 | 4.761 ± 0.51 | 0.703 ± 0.04 | 0.146 ± 0.01 |
| UNIGUIDE | 7.0 | 10/50 | -7.076 ± 2.27 | -9.038 ± 1.38 | 0.550 ± 0.18 | 0.579 ± 0.14 | 4.767 ± 0.50 | 0.704 ± 0.04 | 0.131 ± 0.01 |

For all tables, we conduct the experiments both with and without resampling. The VINA Dock score is measured with QuickVina2 [72], available under the Apache License, and the chemical properties (QED, SA, Lipinski) are measured with RDKit. We note that in all ablation tables we measure the VINA Dock score on the processed molecules, following Schneuing et al. [11], while the VINA Dock score in Tab. 2 is measured following Guan et al. [67]. Both the VINA Dock score and chemical properties improve with additional resampling steps ($R = 10, T = 50$) for both datasets. Additionally, increasing the guidance scale improves the RMSD with respect to the target protein, and results in generating ligands with an improved binding affinity (lower VINA).

## E.6   Additional Results for SBDD

Supplementary to Tab. 2 we provide additional metrics for the evaluation of the generated ligands in Tab. 14: the validity as measured by RDKit [73] and the connectivity, representing the percentage of valid molecules without any disconnected fragments. Additionally, we report the uniqueness and novelty of the valid connected ligands.

## E.7   Runtime Comparison

In Tab. 13, we provide a comparison of the different controlled generation mechanisms regarding their runtime. While UniGuide has a higher runtime compared to other conditioning mechanisms, as it has to compute gradients through the diffusion model at inference time, it stays comparable to other mechanisms such as inpainting.

Table 10: Results for the Binding MOAD test set with the unconditional DiffSBDD base model trained on the full-atom context of the pockets combined with UniGuide and the inpainting-inspired technique DiffSBDD [11]. We provide results for varying the guidance scales $S$ during our controlled generation. We also report results for the conditional diffusion model DiffSBDD-cond.

| METHOD | $S$ | $R/T$ | VINA (↓) | VINA TOP 10% (↓) | QED (↑) | SA (↑) | LIPINSKI (↑) | DIVERSITY (↑) | RMSD (↓) |
|---|---|---|---|---|---|---|---|---|---|
| DIFFSBDD-COND | - | - | -7.172 ± 1.88 | -9.174 ± 2.13 | 0.430 ± 0.20 | 0.564 ± 0.12 | 4.526 ± 0.80 | 0.711 ± 0.08 | 0.0 ± 0.0 |
| DIFFSBDD | - | 1/500 | -6.540 ± 2.00 | -8.427 ± 1.39 | 0.413 ± 0.20 | 0.531 ± 0.11 | 4.611 ± 0.77 | 0.748 ± 0.03 | 55 ± 31 |
| UNIGUIDE | 6.0 | 1/500 | -6.696 ± 1.78 | -8.561 ± 1.58 | 0.407 ± 0.19 | 0.527 ± 0.11 | 4.587 ± 0.78 | 0.740 ± 0.04 | 55 ± 31 |
| UNIGUIDE | 7.0 | 1/500 | -6.683 ± 1.91 | -8.575 ± 1.52 | 0.406 ± 0.19 | 0.524 ± 0.11 | 4.579 ± 0.80 | 0.738 ± 0.04 | 55 ± 31 |
| UNIGUIDE | 8.0 | 1/500 | -6.682 ± 1.77 | -8.555 ± 1.52 | 0.407 ± 0.19 | 0.526 ± 0.11 | 4.591 ± 0.78 | 0.740 ± 0.04 | 55 ± 31 |
| UNIGUIDE | 9.0 | 1/500 | -6.689 ± 1.74 | -8.541 ± 1.50 | 0.403 ± 0.19 | 0.524 ± 0.11 | 4.589 ± 0.78 | 0.738 ± 0.04 | 55 ± 31 |
| DIFFSBDD | - | 10/50 | -7.263 ± 4.19 | -9.776 ± 2.25 | 0.546 ± 0.21 | 0.618 ± 0.12 | 4.777 ± 0.54 | 0.740 ± 0.05 | 53 ± 31 |
| UNIGUIDE | 5.0 | 10/50 | -7.470 ± 2.97 | -9.621 ± 1.84 | 0.563 ± 0.20 | 0.605 ± 0.12 | 4.807 ± 0.50 | 0.723 ± 0.05 | 55 ± 31 |
| UNIGUIDE | 6.0 | 10/50 | -7.570 ± 3.20 | -9.731 ± 1.90 | 0.566 ± 0.20 | 0.606 ± 0.12 | 4.815 ± 0.48 | 0.722 ± 0.05 | 55 ± 31 |
| UNIGUIDE | 7.0 | 10/50 | -7.639 ± 2.39 | -9.793 ± 2.06 | 0.559 ± 0.20 | 0.605 ± 0.12 | 4.804 ± 0.49 | 0.723 ± 0.05 | 54 ± 31 |
| UNIGUIDE | 8.0 | 10/50 | -7.635 ± 2.71 | -9.821 ± 2.07 | 0.558 ± 0.20 | 0.605 ± 0.12 | 4.804 ± 0.50 | 0.720 ± 0.05 | 54 ± 31 |
| UNIGUIDE | 9.0 | 10/50 | -7.661 ± 2.99 | -9.864 ± 2.13 | 0.556 ± 0.20 | 0.605 ± 0.12 | 4.799 ± 0.50 | 0.723 ± 0.05 | 55 ± 31 |

Table 11: Evaluation of the samples generated for the CrossDocked test set using the joint ligand-protein diffusion model trained on the $C_\alpha$ pocket representation for varying guidance scales $S$. The base model is combined either with the inpaitning-inspired technique (DiffSBDD) or UniGuide. We further report the evaluation of the molecules generated by the conditional model DiffSBDD-cond that is trained on the $C_\alpha$ pocket representation.

| METHOD | $S$ | $R/T$ | VINA (↓) | VINA TOP 10% (↓) | QED (↑) | SA (↑) | LIPINSKI (↑) | DIVERSITY (↑) | RMSD (↓) |
|---|---|---|---|---|---|---|---|---|---|
| DIFFSBDD-COND ($C_\alpha$) | - | - | -6.770 ± 2.73 | -8.796 ± 1.75 | 0.475 ± 0.22 | 0.612 ± 0.12 | 4.536 ± 0.91 | 0.725 ± 0.06 | 0.000 ± 0.00 |
| DIFFSBDD | - | 1/500 | -6.485 ± 2.50 | -8.472 ± 1.62 | 0.510 ± 0.21 | 0.619 ± 0.12 | 4.640 ± 0.73 | 0.735 ± 0.06 | 0.053 ± 0.03 |
| UNIGUIDE | 2.0 | 1/500 | -6.528 ± 2.64 | -8.527 ± 1.67 | 0.518 ± 0.21 | 0.623 ± 0.12 | 4.649 ± 0.73 | 0.739 ± 0.05 | 0.085 ± 0.01 |
| UNIGUIDE | 3.0 | 1/500 | -6.604 ± 2.57 | -8.556 ± 1.64 | 0.519 ± 0.21 | 0.622 ± 0.12 | 4.657 ± 0.72 | 0.738 ± 0.05 | 0.070 ± 0.01 |
| UNIGUIDE | 4.0 | 1/500 | -6.578 ± 2.72 | -8.563 ± 1.68 | 0.518 ± 0.21 | 0.623 ± 0.12 | 4.659 ± 0.71 | 0.741 ± 0.05 | 0.059 ± 0.02 |
| UNIGUIDE | 5.0 | 1/500 | -6.563 ± 2.58 | -8.549 ± 1.66 | 0.516 ± 0.21 | 0.624 ± 0.12 | 4.646 ± 0.72 | 0.741 ± 0.05 | 0.052 ± 0.01 |
| UNIGUIDE | 6.0 | 1/500 | -6.658 ± 2.50 | -8.578 ± 1.69 | 0.527 ± 0.21 | 0.629 ± 0.12 | 4.683 ± 0.69 | 0.741 ± 0.05 | 0.045 ± 0.01 |
| DIFFSBDD | - | 10/50 | -7.030 ± 3.39 | -9.057 ± 1.79 | 0.559 ± 0.21 | 0.730 ± 0.12 | 4.729 ± 0.60 | 0.720 ± 0.07 | 0.052 ± 0.01 |
| UNIGUIDE | 1.0 | 10/50 | -6.909 ± 3.35 | -9.069 ± 1.79 | 0.563 ± 0.21 | 0.734 ± 0.12 | 4.743 ± 0.57 | 0.721 ± 0.06 | 0.711 ± 0.12 |
| UNIGUIDE | 2.0 | 10/50 | -7.015 ± 3.20 | -9.115 ± 1.79 | 0.562 ± 0.21 | 0.733 ± 0.12 | 4.735 ± 0.60 | 0.721 ± 0.07 | 0.188 ± 0.02 |
| UNIGUIDE | 3.0 | 10/50 | -7.081 ± 2.95 | -9.140 ± 1.83 | 0.560 ± 0.20 | 0.732 ± 0.11 | 4.742 ± 0.57 | 0.723 ± 0.07 | 0.127 ± 0.01 |
| UNIGUIDE | 4.0 | 10/50 | -7.086 ± 3.27 | -9.125 ± 1.81 | 0.561 ± 0.19 | 0.731 ± 0.10 | 4.729 ± 0.60 | 0.719 ± 0.06 | 0.102 ± 0.01 |
| UNIGUIDE | 5.0 | 10/50 | -7.117 ± 2.78 | -9.127 ± 1.78 | 0.561 ± 0.20 | 0.731 ± 0.12 | 4.738 ± 0.59 | 0.722 ± 0.07 | 0.090 ± 0.01 |
| UNIGUIDE | 6.0 | 10/50 | -7.113 ± 3.00 | -9.133 ± 1.80 | 0.556 ± 0.20 | 0.731 ± 0.12 | 4.734 ± 0.60 | 0.720 ± 0.32 | 0.077 ± 0.01 |

# F  Fragment-based drug design

## F.1  Linker Design

For the experimental evaluation of the linker design task, we follow Igashov et al. [13], employ the ZINC dataset [74] and preprocess it following Igashov et al. [13]. That is, 3D conformers are generated from the SMILES strings present in the dataset with RDKit [73]. We fragment the dataset ligands using an MMPA-based algorithm [75, 73], generating multiple fragment conditions per molecule. We train an unconditional EDM model for this task as specified in App. C. For the evaluation metrics, we follow Igashov et al. [13]. Note that the synthetic accessibility score computation (SA) in Tab. 3 differs from the remaining experimental evaluations. While Igashov et al. [13] report the SA score $s_{SA}$ directly, Schneuing et al. [11] report the SA score as $(10 - s_{SA})/9$.

For the task of linker design, we adjust the condition map as discussed in Sec. 4.2 slightly to include anchor information, similar in spirit to the DiffLinker model incorporating anchor information [13]. That is, additionally to guiding parts of the molecule to the desired fragment configuration, we additionally define a cuboid's surface that is defined from the specified anchor atoms. We can then utilise this surface condition $C_{\partial V}$ to guide the linker atoms in accordance with Eq. (21). Additionally, we can expand this surface based on the linker size to ensure chemical validity of the generated linker. This condition map highlights the flexibility of UniGuide condition maps in various tasks, especially through the combination of two definitions of the condition map. For the experimental evaluation, we sample the size of the linker nodes uniformly in accordance with Igashov et al. [13] and compare to the DiffLinker model without an external network to predict the linker size. Note, however, that also the unconditional EDM model combined with UniGuide can be adapted to include such predictors.

Table 12: Results for the CrossDocked test set with the joint model trained on the full-atom pocket representation of the pocket for varying guidance scales $S$. The unconditional model is either controlled by the inpainting-inspired technique (DiffSBDD) or UniGuide.

| METHOD | $S$ | $R/T$ | VINA (↓) | VINA TOP 10% (↓) | QED (↑) | SA (↑) | LIPINSKI (↑) | DIVERSITY (↑) | RMSD (↓) |
|---|---|---|---|---|---|---|---|---|---|
| DIFFSBDD-COND | - | - | -6.950 ± 2.06 | -9.120 ± 2.16 | 0.469 ± 0.21 | 0.578 ± 0.13 | 4.562 ± 0.89 | 0.728 ± 0.07 | 0.000 ± 0.00 |
| DIFFSBDD | - | 1/500 | -6.225 ± 1.77 | -8.115 ± 1.64 | 0.469 ± 0.20 | 0.573 ± 0.11 | 4.691 ± 0.70 | 0.778 ± 0.04 | 0.049 ± 0.01 |
| UNIGUIDE | 5.0 | 1/500 | -6.346 ± 1.74 | -8.208 ± 1.62 | 0.482 ± 0.20 | 0.570 ± 0.12 | 4.718 ± 0.67 | 0.773 ± 0.04 | 0.040 ± 0.01 |
| UNIGUIDE | 6.0 | 1/500 | -6.335 ± 1.72 | -8.225 ± 1.61 | 0.484 ± 0.20 | 0.571 ± 0.12 | 4.715 ± 0.66 | 0.775 ± 0.04 | 0.039 ± 0.01 |
| UNIGUIDE | 7.0 | 1/500 | -6.338 ± 1.73 | -8.218 ± 1.60 | 0.481 ± 0.19 | 0.571 ± 0.12 | 4.710 ± 0.67 | 0.774 ± 0.04 | 0.039 ± 0.01 |
| UNIGUIDE | 8.0 | 1/500 | -6.366 ± 1.72 | -8.261 ± 1.57 | 0.485 ± 0.20 | 0.570 ± 0.12 | 4.717 ± 0.66 | 0.773 ± 0.03 | 0.039 ± 0.01 |
| DIFFSBDD | - | 10/50 | -7.216 ± 2.54 | -9.490 ± 2.00 | 0.571 ± 0.19 | 0.639 ± 0.14 | 4.808 ± 0.50 | 0.707 ± 0.09 | 0.045 ± 0.01 |
| UNIGUIDE | 6.0 | 10/50 | -7.295 ± 2.22 | -9.441 ± 1.95 | 0.574 ± 0.19 | 0.641 ± 0.14 | 4.825 ± 0.47 | 0.706 ± 0.08 | 0.047 ± 0.01 |
| UNIGUIDE | 7.0 | 10/50 | -7.320 ± 2.27 | -9.514 ± 2.04 | 0.571 ± 0.19 | 0.638 ± 0.14 | 4.822 ± 0.47 | 0.705 ± 0.08 | 0.047 ± 0.01 |
| UNIGUIDE | 8.0 | 10/50 | -7.298 ± 2.21 | -9.460 ± 2.01 | 0.568 ± 0.19 | 0.641 ± 0.14 | 4.818 ± 0.47 | 0.703 ± 0.09 | 0.048 ± 0.01 |
| UNIGUIDE | 9.0 | 10/50 | -7.265 ± 2.45 | -9.495 ± 2.05 | 0.577 ± 0.19 | 0.640 ± 0.14 | 4.821 ± 0.47 | 0.706 ± 0.08 | 0.049 ± 0.01 |

Table 13: We evaluate the runtime of UniGuide and compare it to DiffSBDD-cond and DiffSBDD from Schneuing et al. [11]. We report the average time (in seconds) to generate 100 ligands per pocket for the CrossDocked ($C_\alpha$), Binding Moad ($C_\alpha$) and Binding Moad (fullatom).

| DATASET | MODEL | RUNTIME (S) |
|---|---|---|
| CROSSDOCKED ($C_\alpha$) | DIFFSBDD-COND | $60 \pm 68$ |
| | DIFFSBDD | $141 \pm 55$ |
| | UNIGUIDE | $193 \pm 61$ |
| BINDING MOAD ($C_\alpha$) | DIFFSBDD-COND | $54 \pm 42$ |
| | DIFFSBDD | $61 \pm 17$ |
| | UNIGUIDE | $104 \pm 36$ |
| BINDING MOAD (FULL) | DIFFSBDD-COND | $345 \pm 55$ |
| | DIFFSBDD | $398 \pm 95$ |
| | UNIGUIDE | $453 \pm 120$ |

Table 14: Additional metrics for the methods discussed in Sec. 5.2.

| | | VALIDITY (↑) | CONNECTIVITY (↑) | UNIQUENESS (↑) | NOVELTY (↑) |
|---|---|---|---|---|---|
| | TEST SET | 100% | 100% | 96.00% | 96.88% |
| CROSS-DOCKED | DIFFSBDD-COND ($C_\alpha$) | 95.32% | 80.63% | 99.97% | 99.81% |
| | DIFFSBDD-COND | 97.32% | 78.91% | 99.99% | 99.91% |
| | DIFFSBDD ($C_\alpha$) | 99.20% | 98.14% | 99.26% | 99.16% |
| | DIFFSBDD | 97.76% | 89.84% | 99.94% | 99.87% |
| | UNIGUIDE ($C_\alpha$) | 99.12% | 98.35% | 99.50% | 99.24% |
| | UNIGUIDE | 97.40% | 93.18% | 99.93% | 99.76% |
| | TEST SET | 97.69% | 100% | 38.58% | 77.55% |
| BINDING MOAD | DIFFSBDD-COND ($C_\alpha$) | 94.43% | 77.17% | 100% | 100% |
| | DIFFSBDD-COND | 96.20% | 63.20% | 100% | 100% |
| | DIFFSBDD ($C_\alpha$) | 98.54% | 91.45% | 100% | 100% |
| | DIFFSBDD | 94.22% | 75.60% | 100% | 100% |
| | UNIGUIDE ($C_\alpha$) | 98.44% | 93.12% | 100% | 99.99% |
| | UNIGUIDE | 93.85% | 79.95% | 100% | 100% |

## F.2 General Fragment Conditions

To assess the performance of UniGuide for the task of FBDD, we create an experimental setup with the goal of generating ligands conditioned on desired fragments roughly following [13]. We select 10 random protein targets from the Binding MOAD dataset and decompose their corresponding reference ligands using an MMPA-based algorithm [75, 73]. This decomposition results in a set of 40 different scenarios, including separated fragments we want to link, a fragment to grow or small functional groups to perform scaffolding. For every set of fixed fragments, we aim to guide the unconditional generation of ligands towards the generation of a ligand containing the desired fragments. As the protein is not the target of the guidance, we employ the DiffSBDD-cond model, which is conditionally trained on the ($C_\alpha$)-representation of the protein pocket. For every set of fixed fragments, we generate 100 ligands and use a constant guidance scale of 8.

We provide quantitative results for the task of fragment-based drug design in Tab. 15. On the one hand, the task requires the desired fragments to be present in the generated molecule. Thus, we measure the success rate of recovery (Hit Ratio) and the RMSD between the generated fragments and desired fragments. On the other hand, given that the target fragments are met in the generated ligand, the generation has to achieve favourable chemical properties, high binding affinity, as well as high diversity within the set of generated ligands and low similarity to the reference ligand. As the Inpaint mechanism enforces the fragment during generation more strictly, it is able to achieve a better Hit Ratio and RMSD. Nevertheless, UniGuide achieves competitive results but also better VINA docking scores, better properties, and lower similarity compared to the reference ligand.

The FBDD task puts a hard constraint on the generated ligands, namely that a set of desired fragments has to be present in the generated ligand. However, neither DiffSBDD nor UniGuide can guarantee that the condition fragments are present in the generated samples.

We provide further qualitative results of the generated ligands for the FBDD task in Fig. 7.

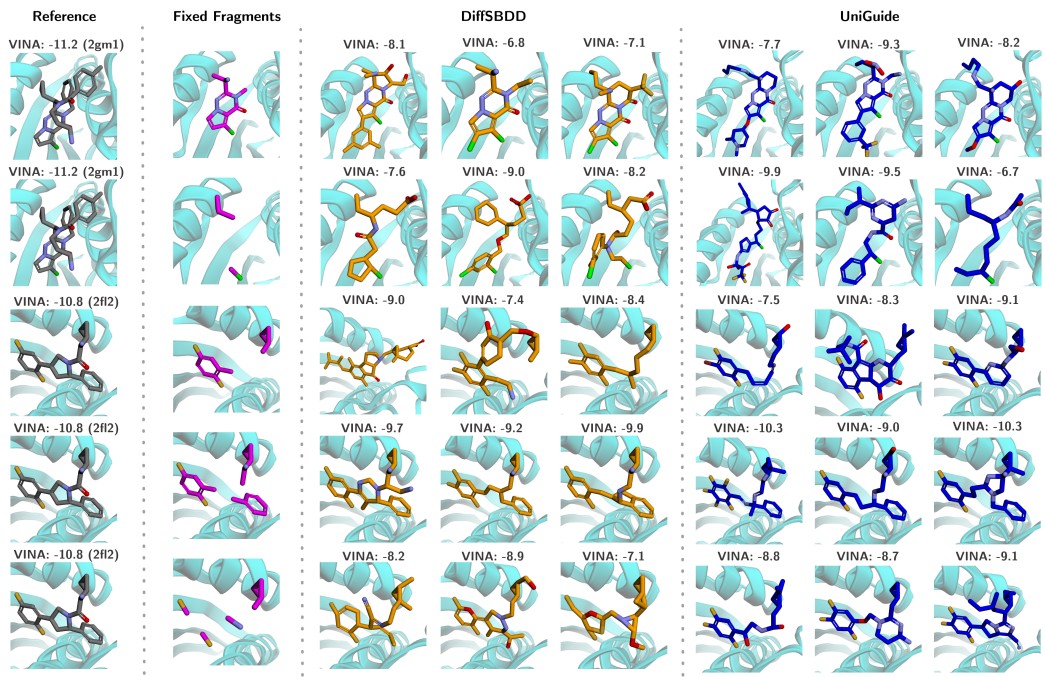

Figure 7: Examples of the generated fragment conditioned ligands.

Table 15: Quantitative comparison between DiffSBDD and UniGuide for the FBDD task on the Binding MOAD ($C_\alpha$) dataset. As the condition in this FBDD scenario is a hard constraint that entails the condition to be exactly present in the generation, we add a post-hoc step for both methods where we replace the inpainted or guided parts with the exact condition atoms. We report mean and standard deviation and highlight the best method in **bold**.

|  | DIFFSBDD | UNIGUIDE |
|---|---|---|
| VINA ($\downarrow$) | -7.406 $\pm$ 0.79 | **-7.924** $\pm$ **0.89** |
| QED ($\uparrow$) | 0.612 $\pm$ 0.11 | **0.639** $\pm$ **0.09** |
| SA ($\uparrow$) | **0.703** $\pm$ **0.11** | 0.691 $\pm$ 0.10 |
| LIPINSKI ($\uparrow$) | 4.819 $\pm$ 0.28 | **4.875** $\pm$ **0.19** |
| DIVERSITY ($\uparrow$) | 0.653 $\pm$ 0.28 | **0.669** $\pm$ **0.23** |
| SIMILARITY ($\downarrow$) | **0.172** $\pm$ **0.02** | 0.177 $\pm$ 0.02 |
| VALIDITY ($\uparrow$) | 93.35 % | **94.41** % |
| CONNECTIVITY ($\uparrow$) | 66.87 % | **68.30** % |

# G  Atom densities in 3D space

Similar to the guidance by the volume enclosed by the molecular surface, UniGuide allows to guide towards multiple point clouds simultaneously. A natural extension of LBDD would be to harness atom densities as described in Zaucha et al. [82]. Such a setting combines aspects of LBDD and SBDD as it provides conditions also on the feature space, yet the source can only be represented by point clouds.

In particular, we anticipate UniGuide to be useful in scenarios where explicit information about advantageous features of the ligand is provided in the form of 3D densities. Examples of this include a) volumetric densities that indicate beneficial placement of certain atom types, such as oxygen atoms [82] or b) pharmacophore-like retrieval of advantageous positions for aromatic rings, as utilised in e.g. Zhu et al. [83]. On a technical level, this setting assumes that instead of a reference ligand's structure, we only have access to (multiple) atom type densities that indicate preferred locations for optimal interaction with the protein. Additionally, instead of conditioning on a reference ligand's shape, we could condition on a protein pocket's surface, which primarily defines exclusion zones rather than precise atom placement.

Adapting UniGuide for such scenarios requires only minor adjustments, as the protein surface can treated like shapes in standard LBDD, defining an exclusion zone based on proximity to the surface. The atom densities are thresholded to reflect regions of high interest and converted to surfaces using the marching cubes algorithm [84]. To also include feature information, we effectively employ a modified condition map similar to Eq. (21) that extends the transformation from the conformation to the configuration space. Moreover, the number of atoms guided by each density is adjusted based on its volume, reflecting the varying influence of each density, and guidance is only applied if atoms are sufficiently close.

We show explorative results for the guided generation of molecules towards desired atom densities using UniGuide in Fig. 8. While our current approach represents a promising first step in tackling this task, we acknowledge the potential for further refinement and are eager to explore future improvements within the UniGuide framework.

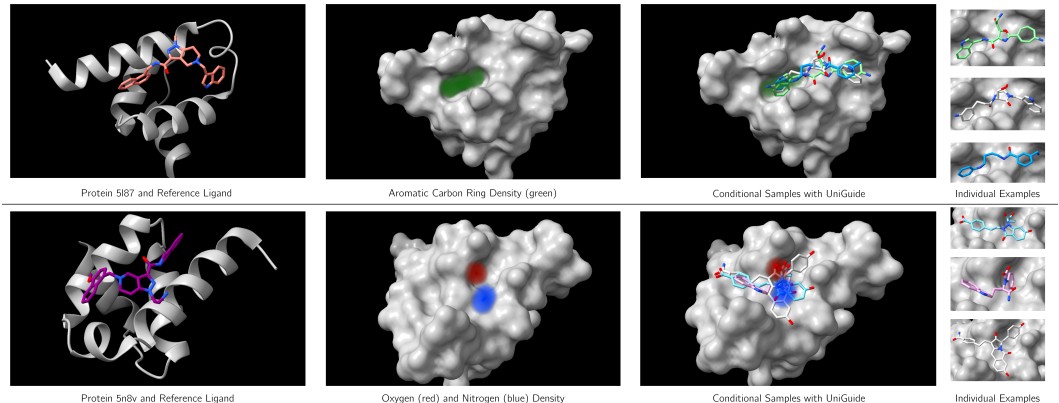

Figure 8: Given a source density of oxygens, we can extend UniGuide to generate ligands satisfying the condition.

