# OpenReview forum: "Unified Guidance for Geometry-Conditioned Molecular Generation"
_NeurIPS.cc/2024/Conference — NeurIPS 2024 poster_

### Official Review · Reviewer_D4nw · 2024-07-02

**Soundness:** 4
**Presentation:** 3
**Contribution:** 3
**Rating:** 7
**Confidence:** 4

**Summary:**

The paper introduces UniGuide, a unified framework for geometry-conditioned molecular generation using unconditional diffusion models. UniGuide is designed to address the adaptability issues in current molecular diffusion models by providing a general training-free approach via a condition map that transforms complex geometric conditions to match the diffusion model’s configuration space, allowing for self-guidance during the generation process. UniGuide is demonstrated to be effective in various drug discovery tasks, including structure-based, fragment-based, and ligand-based drug design. The framework shows either on-par or superior performance compared to specialized models, highlighting its potential to streamline the development of molecular generative models.

**Strengths:**

1. The paper introduces a novel, training-free method to guide diffusion models based on expected geometry conditions, enhancing adaptability without additional training overhead.
2. The framework is thoroughly evaluated across multiple drug discovery tasks, demonstrating its effectiveness.

**Weaknesses:**

1. The contribution and the generalizability of UniGuide are somewhat overstated. In the Introduction and Figure 1, the available conditions are described in a quite general way, including not only structures and surfaces, but also densities. As this paper mainly focuses on geometry-aware conditions, clarifying the current scope in the introduction and main figures would enhance the paper's accuracy.

**Questions:**

1. In the FBDD experiments, can the UniGuide framework also be applied to the DiffLinker model, considering it is diffusion-based backbone?
2. Based on W1, it is noticed that there is a brief exploration on the density condition in App. G. Given its potential to enhance the model's capabilities, it is suggested to provide additional details about this setting (e.g., task definition, condition mapping) and to reference App. G in the main body of the paper.

**Limitations:**

The limitation of this paper is well discussed

---

> ### Author Rebuttal · Authors · 2024-08-07
>
> We are grateful for the reviewer's insightful feedback and are pleased with the positive comments on UniGuide's novelty and effectiveness for various drug discovery tasks. We would like to clarify the remaining questions and concerns in the following.
>
> &nbsp;
>
> > The contribution and the generalizability of UniGuide are somewhat overstated; the available conditions are described in a quite general way, including not only structures and surfaces, but also densities
>
> We appreciate the reviewer's feedback and acknowledge the ambiguity between UniGuide's contributions, as shown in Fig. 1, and the empirical evaluations presented in our experimental section. To resolve this ambiguity:
> - We **extended the setting from App.G on density-conditioned guidance as part of our rebuttal** and explain it in more detail down below.
> - We plan to incorporate this novel setting into the final part of Sec.5.1 in the updated manuscript, referring to it as density-based drug design (DBDD), a more challenging task than LBDD.
>
> > Given its potential [of guiding w.r.t. densities] to enhance the model's capabilities, it is suggested to provide additional details about this setting (e.g., task definition, condition mapping) and to reference App. G in the main body of the paper
>
> We appreciate the reviewer's suggestion and are pleased to provide a more detailed explanation of the technical aspects of the DBDD setting.
>
> Task Definition
> - As motivated in App.G, we anticipate UniGuide to be particularly useful in scenarios where explicit information about advantageous features of the ligand is provided in the form of 3D densities. Examples of this include
>     - a) volumetric densities that indicate beneficial placement of certain atom types, such as oxygen atoms [2] or
>     - b) pharmacophore-like retrieval of advantageous positions for aromatic rings, as utilised in [3].
> - On a technical level, the DBDD setting assumes this information to be provided as follows:
>     - Instead of a reference ligand's shape, we only have the protein pocket's surface, which primarily defines exclusion zones rather than precise atom placement. Please refer to Fig.1 in the attached PDF.
>     - Instead of a reference ligand's structure, we only have access to (multiple) atom-type densities that indicate preferred locations for optimal interaction with the protein.
>
> Condition Map: Adapting UniGuide for the DBDD scenario requires minor adjustments
> - The protein surface is treated like shapes in standard LBDD, defining an exclusion zone based on proximity to the surface.
> - The atom densities are thresholded to reflect regions of high interest and converted to surfaces using the marching cubes algorithm [4]
> - To include feature information, we employ a modified condition map similar to Eq.21 that extends the transformation from the conformation to the configuration space.
> - Moreover, the number of atoms guided by each density is adjusted based on its volume, reflecting the varying influence of each density, and guidance is only applied if atoms are sufficiently close.
>
> **We have included a new figure in the attached PDF that visually demonstrates qualitative results for this setting**. While our current approach represents a promising first step in tackling this task, we acknowledge the potential for further refinement. We are eager to explore future improvements within the UniGuide framework.
>
> > For FBDD [...], can the UniGuide framework also be applied to the DiffLinker model, considering it is diffusion-based backbone
>
> While direct application of UniGuide's FBDD condition map is not possible with DiffLinker, it can be combined with the surface-based Linker Design condition map (App.F.1) to guide linker atom positions.
> - DiffLinker's Modifications: Since DiffLinker fixes condition fragments in space and only learns to diffuse the linker atoms, the model becomes unsuitable for UniGuide's FBDD condition map from Sec. 4.2.
> - DiffLinker's Similarity to EDM: However, DiffLinker's diffusion backbone is very similar to EDM, which we successfully combined with UniGuide for controlled generation in Linker Design, see Tab.3.
> - Compatibility with Surface-Based Conditioning: Moreover, DiffLinker can still be combined with the surface-based Linker Design condition map (App.F.1) to guide the placement of linker atoms towards a specific region between the fragments.
>
> &nbsp;
>
> We appreciate the positive feedback and are excited to move forward with these improvements. We hope that we have addressed all outstanding concerns satisfactorily and welcome further discussion.
>
> &nbsp;
>
> [1] Hoogeboom, Emiel, et al. Equivariant diffusion for molecule generation in 3d, 2022.
>
> [2] Zaucha, Jan, et al. Deep learning model predicts water interaction sites on the surface of proteins using limited-resolution data, 2020
>
> [3] Zhu, Huimin, et al. A pharmacophore-guided deep learning approach for bioactive molecular generation, 2023
>
> [4] Lorensen, William E., and Harvey E. Cline. Marching cubes: A high resolution 3D surface construction algorithm, 1998

---

> > ### Comment · Reviewer_D4nw · 2024-08-08
> >
> > Thanks for your explanations and additional results. My concern has been resolved and I have raised my score to 7.

---

> > > ### Author Response · Authors · 2024-08-13
> > > **Thank you**
> > >
> > > Thank you very much for the kind words and for increasing the score of our paper. We are pleased to hear that we were able to resolve the reviewer's concerns.
> > >
> > > Best regards,
> > >
> > > The Authors

---

### Official Review · Reviewer_YA7u · 2024-07-09

**Soundness:** 2
**Presentation:** 2
**Contribution:** 2
**Rating:** 4
**Confidence:** 3

**Summary:**

This paper proposed a training-free framework for guided diffusions in unconditional molecular generation. UniGuide applies to a wide range of design tasks such as SBDD, FBDD and LBDD, unified by the proposed condition map, which projects from the product space of conditional input that lies in general geometric space to some favorable datapoint in the data space that diffusion operates within.

**Strengths:**

- The idea of condition map that unifies different downstream tasks is novel.
- This paper is generally easy to follow.

**Weaknesses:**

- In all tables, the authors claimed to highlight the best "diffusion-based approach" in bold, which seems misleading since there are also non-diffusion baselines. I would recommend the authors to reconsider this style of presentation in order to faithfully reveal the general performance.
- After checking other baseline results, UniGuide seems not so competitive in downstream tasks such as LBDD and linker design in FBDD. This casts doubt on its effectiveness.
- There are a number of important baselines missing, e.g. DecompDiff [1] and IPDiff [2] for SBDD tasks, and LinkerNet [3] for LBDD.

[1] DecompDiff: Diffusion Models with Decomposed Priors for Structure-Based Drug Design

[2] Protein-Ligand Interaction Prior for Binding-aware 3D Molecule Diffusion Models

[3] LinkerNet: Fragment Poses and Linker Co-Design with 3D Equivariant Diffusion

**Questions:**

- The condition map works similarly to molecular translation or some form of retrieval given conditional inputs. How would the authors compare to those methods?
- For SBDD, the authors only reported QVina Dock scores in the main results. However, SBDD models have been criticized for inaccurate structure modeling [1]. It seems to me that Vina Score and Vina Minimize used in [2] would serve as a better indicator for the pose quality. Can the authors also include these metrics in their results?

[1] Benchmarking Generated Poses: How Rational is Structure-based Drug Design with Generative Models?

[2] DecompDiff: Diffusion Models with Decomposed Priors for Structure-Based Drug Design

**Limitations:**

See above.

---

> ### Author Rebuttal · Authors · 2024-08-07
>
> We thank the reviewer for their constructive feedback regarding our manuscript, particularly regarding the presentation and interpretation of UniGuide. We believe we can address these concerns effectively, as detailed in the following answers.
>
> &nbsp;
>
> > Only highlight the best diffusion-based approach
> - We are happy to revise the updated manuscript's presentation style as the reviewer suggested.
> - We want to emphasise that we never intended to mislead the reader by highlighting the results in a way that favours UniGuide. We are confident in the benefits provided by UniGuide and agree that they should be presented without ambiguity.
>
> > UniGuide seems not so competitive in LBDD and Linker design in FBDD tasks
> - Clarifications on LBDD performance
>     - We stress that "on-par" performance for UniGuide compared to specialised approaches is a significant achievement, as our method imposes no constraints or additional training. The general response provides clarifications on LBDD performance and the reasoning behind the experimental settings. We highlight that **UniGuide achieves very high shape similarity and low graph similarity**, even though it does not rely directly on the reference structure (as in SQUID).
> - Clarifications on FBDD (Linker design) performance
>     - We agree that UniGuide performs on par with DiffLinker overall in the linker design task and does not strictly outperform it. Yet, we want to emphasise again that **we simply guide the sampling of an EDM [1] model that was not optimised for linker design** (unlike DiffLinker, which is a conditional model limited to the linker design task only). In Fig.4 and App. F.2, we showcase how UniGuide can be flexibly adapted to more tasks such as scaffolding or fragment growing.
>     - Despite not being trained explicitly for linker design, UniGuide(EDM) performs competitively with DiffLinker: Our method generates the most diverse linkers (Uniqueness) and successfully recovered nearly 60% (Recovery) of the actual linkers from the test set. While baseline methods may achieve higher recovery rates, they produce less diverse solutions. Moreover, UniGuide (EDM) generates more complex structures as measured by the number of rings. However, this also leads to a slightly worse SA score as these molecules are not as easily synthesisable.
>     - Another advantage of UniGuide not relying on extra training for the linker design task is that it is **agnostic to the fragmentation procedure used to obtain the condition fragments**. This means that UniGuide will generalise to unseen fragments if the underlying molecule fits within the training distribution.
>     - We realise that the original manuscript lacked clarity on these aspects, which may lead to misunderstandings. In the updated version, we will better contextualise the FBDD results.
>
> > Additional baselines
> - SBDD:
>     - We include DecompDiff and IPDiff in the updated version of the manuscript as requested by the reviewer. Furthermore, we are happy to include additional Vina metrics to indicate the pose quality better. Please refer to Tab.1 in the attached PDF. We have also provided additional experimental details and comparisons to different baselines in the general response.
> - FBDD:
>     - We thank the reviewer for pointing us towards the LinkerNet work. Like DiffLinker, LinkerNet is a conditional model that is additionally able to model fragments with more degrees of freedom, i.e. their pose and centre of mass. Since this is also reflected in their experimental setup (LinkerNet can predict the condition fragment poses, the baselines randomly rotate them), it is not directly comparable to the experiments conducted in Table 3. However, this is an exciting addition to the linker design task, which can be readily implemented with minor adjustments to the condition maps presented by UniGuide. We will investigate this addition for inclusion in the updated version of the manuscript.
>
> > Q: Connection to retrieval and molecular translation
> - Comparison to molecular translation: While UniGuide's primary focus is novel drug design with geometric constraints, its adaptability allows for future exploration of molecular translation in 3D, similar to existing 2D approaches [3,4]. This could include, for example, optimising a molecule's properties while preserving its 3D shape using the surface condition map. However, this would necessitate property regressors and the combination of UniGuide with classifier guidance.
> - Comparison to retrieval-based methods: Retrieval-based methods, such as shape-based Virtual Screening (VS),  involve retrieving existing data based on predefined criteria or queries, whereas UniGuide focuses on generating new samples based on a predefined condition without involving existing data. Our comparisons for LBDD show that we surpass VS in terms of 3D and 2D similarity (Tab.1). We are curious to explore the inclusion of retrieval-based methods into the condition map in the future.
>
> &nbsp;
>
> We hope that we have adequately addressed the concerns raised and believe the reviewer's feedback has significantly enhanced the presentation of our results and the experimental section. We look forward to further discussion.
>
> &nbsp;
>
> [1] Hoogeboom et al. Equivariant diffusion for molecule generation in 3d
>
> [2] Schneuing et al. Structure-based drug design with equivariant diffusion models
>
> [3] Jin et al. Hierarchical generation of molecular graphs using structural motifs
>
> [4] Jin et al. Junction tree variational autoencoder for molecular graph generation

---

> > ### Comment · Reviewer_YA7u · 2024-08-13
> >
> > Thank you for the response. Given the additional results and comparison with more diffusion baselines, I'm still a bit concerned about the effectiveness and necessity of introducing geometric guidance, since it only marginally improves upon the performance of backbone models (EDM, DiffSBDD, etc). The author mentioned that these models have not been specifically engineered for certain tasks, but it seems to me that the point of adapting existing models for downstream tasks is primarily aimed to benefit from them so as to boost the task-specific performance, which UniGuide has yet to achieve. In this regard, I'm inclined to maintain the score.

---

> > > ### Author Response · Authors · 2024-08-13
> > > **Clarification on performance and broader impact of UniGuide**
> > >
> > > Thank you very much for your response. While we consider UniGuide's performance improvements significant (details below), we want to emphasise that a performance-focused discussion overlooks UniGuide's **broader potential and impact: UniGuide's formulation enables to tackle entirely novel drug discovery tasks where no established baselines or sufficient data exists.** Specifically, our goal is not to adapt existing models to boost their performance in downstream tasks but to provide a generally applicable guidance framework that makes (unconditional) base models useful for various tasks and practical applications.
> > >
> > > This capability is well demonstrated in our experiments on density-based drug design (no data) and symmetric proteins (limited base model); see Figures in PDF. Our results show that UniGuide is versatile (EDM was applied to both LBDD & FBDD) and effective, consistently delivering performance that matches or exceeds task-specific models (which are limited to a single task and require extra training) and outperforms alternative conditioning mechanisms.
> > >
> > > This sets the basis for UniGuide's long-term objective to reliably translate novel tasks directly to a generative model by incorporating any newly developed geometric conditions through its condition map, thereby accelerating the overall drug discovery process (for real-world applications).
> > >
> > > We also want to highlight a selection of our results that demonstrate the consistency and significance of our improvements, and we kindly ask the reviewer to reassess their evaluation in light of this evidence:
> > >
> > > - **LBDD**: **UniGuide is state-of-the-art** and outperforms alternative guidance mechanisms (cf. follow-up to our general response). An improvement over the base EDM model for LBDD (as indicated by the reviewer) is not possible as **EDM can only be applied for the LBDD tasks with UniGuide** (Table 3, Rebuttal PDF).
> > > - **SBDD**: **UniGuide demonstrates superior performance over all evaluated conditioning mechanisms for diffusion models** (cf. follow-up to our general response). Specifically, we improve the base model DiffSBDD in terms VINA score of up to 1 when utilising UniGuide and up to 1.8 when additionally using UniGuide's version of Clash Drift. This results in competitive performance of the base model compared to conditional, task-specific models such as DecompDiff and IPDiff.
> > > - **FBDD**: **UniGuide enhances the VINA score by 0.5 compared to the evaluated baseline for general, pocket-conditioned FBDD tasks** and generates more valid and connected ligands (as shown in Table 13 of the manuscript).
> > >
> > > While the observed improvement initially appeared marginal to the reviewer, we believe further consideration of the experimental evidence and UniGuide's potential for novel applications and model enhancement may offer a different perspective. We appreciate your feedback and hope our explanation has been helpful in clarifying UniGuide's potential impact.

---

### Official Review · Reviewer_ZcJd · 2024-07-15

**Soundness:** 2
**Presentation:** 3
**Contribution:** 2
**Rating:** 6
**Confidence:** 4

**Summary:**

This paper proposed a method named UniGuide for conditional molecular generation with unconditional diffusion models, without the need of additional training and parameters. The proposed framework is an extension of self-guided diffusion models on the conditional molecular generation task. The authors designed different condition maps C: S x Z $\rightarrow$ Z for ligand-based generation and structure-based / fragment-based generation, enabling guidance from conditions in a unified fashion.

**Strengths:**

* The paper is generally easy to follow and well-written
* The theoretical justification is provided and oblation studies / visualization results are sufficient

**Weaknesses:**

* Technical contribution
    * My biggest concern is on the superiority of the proposed method over other conditional sampling methods. E.g.  the special case of S = Z can be implemented with inpainting technique; the shape-based generation can also be implemented with some technique similar to classifier guidance (e.g. the validity guidance used in [1][2]) by designing a loss function based on denoised datapoint. However, the authors didn't discuss about the superiority of their method with sufficient experimental supports.
    * Limitation: The performance is strongly relied on the based unconditional model.

* Clarification
    * I don’t think the  approximation of Eq 10 makes much sense to me: f approximates the clean data point, then how could the condition c be a Gaussian distribution taking this clean data point as the mean?  I think the concept of condition c $\in$ Z in this paper is closer to datapoint satisfying condition c, or there should be another mapping from the data space to the condition space.
    * Eq 17 doesn’t strictly follow the definition of the condition map:  c and z should have the dimension of n x (3 + d) — Eq (18) should have corresponding index selection / mask operations.

* Experiments
    * In 5.1 Ligand-based drug design, why don’t the authors compare their model with conditional EDM? How could UniGuide (shapeMol), a conditional model with soft constraints, outperform Cond-Shape2Mol?
    * In 5.3 Fragment-based drug design, I think it’d better also compare UniGuide with an inpainting version of other baselines.


References:

[1] Peng, X., Guan, J., Liu, Q., & Ma, J. (2023). Moldiff: Addressing the atom-bond inconsistency problem in 3d molecule diffusion generation. arXiv preprint arXiv:2305.07508.

[2] Guan, J., Zhou, X., Yang, Y., Bao, Y., Peng, J., Ma, J., ... & Gu, Q. (2023). DecompDiff: Diffusion Models with Decomposed Priors for Structure-Based Drug Design. In International Conference on Machine Learning.

**Questions:**

See the weaknesses above.

---

> ### Author Rebuttal · Authors · 2024-08-07
>
> We thank the reviewer for their valuable feedback and will address their concerns below.
>
> &nbsp;
>
> > Superiority of UniGuide and insufficient experimental support
>
> UniGuide's appeal builds upon multiple aspects:
>  - No Training: UniGuide is based on guidance, and as such, it **does not require any additional training but achieves the desired sampling behaviour at inference time**
>  - The unification aspect of UniGuide: **Unconditional models can readily be adapted to new settings via UniGuide**. This adaptation effectively reduces to the definition of a condition map C, which can be non-differentiable. To support this, we guide an unconditional EDM model for the LBDD task (Tab.1) as well as the Linker Design task (Tab.3) and show that we outperform specialised approaches.
>  - The special case of $S=Z$ (not the general case!) can be addressed using techniques based on inpainting. However, UniGuide leads to better or equal performance, making it favourable over inpainting. This is supported across our experiments, such as in Tab.2 or App.F.2.
>  - Extensive experimental support: We benchmark UniGuide against specialised non-diffusion and diffusion-based models (including conditional and guidance-based alternatives) across tasks involving geometric conditions. Our results prove that **UniGuide provides better or on-par performance, indicating that controlling unconditional models with UniGuide is effective, simple, and easily extended to novel settings**. Yet, we acknowledge the need for a more explicit discussion and refer the reviewer to our general response for further clarification on SBDD, LBDD (where UniGuide excels), and FBDD (see also the response to reviewer YA7u).
>
> > Shape-based generation via technique akin to validity guidance
> - We agree that the proposed loss used for validity guidance [1] could be adjusted to the LBDD case, and we thank the reviewer for pointing this out.
> - Yet, **UniGuide is more general and decouples the surface computation (input to the condition map) and gradient computation**, which is consistently applied to the L2 loss between the clean estimate and the target condition.
> - This decoupling has multiple implications:
>     - Due to the loss formulation, the condition map does not have to be differentiable, making it agnostic to how the surface points are computed and, therefore, more flexible.
>     - At the same time, the condition map follows an explicit geometric intuition, which makes it easily adjustable to novel scenarios. This aspect is especially well reflected in the new experiments discussed in our general response. We guide towards a volume reflecting the symmetry requirement to generate symmetric proteins.
> - In our general response, we discuss incorporating drift terms into UniGuide (akin to validity guidance) for SBDD. As part of this addition, we will conduct experiments to compare an adjustment of validity guidance with our surface condition map. We will share the results as soon as they are available.
>
> > Limitation: Reliance on the base model
> - While UniGuide's performance is inherently tied to the performance of the underlying base model (as noted in our limitations), **this connection also has a positive effect as it encourages research on unconditional generation**. Our experiments show that UniGuide can directly translate improvements to better task-specific performance. UniGuide also facilitates the application of these models to diverse downstream tasks, broadening their utility.
>
> > Eq10 does not make much sense
>
> - We apologise for the confusion caused by the poor presentation of Eq.10.
> - In the current manuscript, we missed to specify that the condition $c$ must lie in the same space as the samples $z\_t$ (the configuration space). Only this assumption makes the Gaussian approximation possible. To clarify this, we will add further explanations near Eq.10.
> - We would like to highlight that we discuss and lift this assumption on the condition $c$ in our method section (Sec.4). Note that the condition map $C$ serves as a transformation that takes the source condition $s$ and the clean approximation of $z\_t$ as inputs, and outputs a suitable target condition $c$ that lies in the configuration space and can be used directly in the guidance loss presented in Eq.14/15.
>
> > Eq.17 does not match the definition of C
>
> - In Eq.17, we define the condition map as $C(s, \hat{z}^{\mathcal{A}}\_t)=C(\tilde{z}, \hat{z}^{\mathcal{A}}\_t)$, where $\tilde{z}\in Z$ is a configuration that specifies $m<N$ nodes (for $m=N$ there is no point in guidance).
> - In order to match the dimension of the clean data estimate $\hat{z}\_t$ with $\tilde{z}$, we subset $\hat{z}\_t$ to $m$ nodes as indicated by the superscript $\mathcal{A}$ in the condition map.
> - For example, for SBDD, the source condition is a protein with $m= N^{P}<N$ nodes.
> - While we believe Eq.17 and its definition are consistent, we would appreciate further clarification if we have overlooked any aspects.
>
> > In 5.1 Ligand-based drug design, why do the authors not compare their model with conditional EDM?
>
> In our general response, we clarify our reasoning for the LBDD evaluation and hope that our answer explains why we did not compare UniGuide (EDM) to conditional EDM.
>
> > How could UniGuide (shapeMol), a conditional model with soft constraints, outperform Cond-Shape2Mol?
> - We believe that what is meant by "Cond-Shape2Mol" maps to the conditional ShapeMol [2]. We followed this suggestion and added UniGuide (ShapeMol) to our comparisons; see Tab.3 (PDF).
>
> > For FBDD, add inpainting comparison
> - We added this in Tab.2 in the PDF and would refer to our general response for additional details on this setting.
>
> &nbsp;
>
> Again, we thank the reviewer for their constructive comments, which have significantly improved our work.
>
> &nbsp;
>
>
> [1] Schneuing et al. Structure-based drug design with equivariant diffusion models
>
> [2] Chen et al. Shape-conditioned 3d molecule generation via equivariant diffusion models

---

> > ### Author Response · Authors · 2024-08-13
> > **Additional results posted as official comment**
> >
> > Dear ZcJd,
> >
> > As promised in our initial response, we conducted additional experiments to compare UniGuide with an adaption of validity guidance for LBDD and included a version of "clash drift" into UniGuide's condition map for SBDD. As we believe that the results are relevant to all reviewers, we kindly refer you to our official comment. Thank you again for your valuable feedback.
> >
> > &nbsp;
> >
> > Best regards,
> >
> > The Authors

---

> > > ### Comment · Reviewer_ZcJd · 2024-08-13
> > >
> > > Thank the authors for the detailed response! Most of my concerns have been addressed and I'd love to raise my score to 5. However, the authors still didn't show the superiority and highlight the difference of their method compared to some other techniques like classifier guidance, which doesn't need additional training, can also be applied in different application scenarios, and have very similar mathematical formation. That's the main reason for not giving a higher score.

---

> > > > ### Author Response · Authors · 2024-08-14
> > > > **Highlighting methodological differences and superiority of UniGuide**
> > > >
> > > > &nbsp;
> > > >
> > > > Thank you for your kind answer and for increasing the score. We are glad that our clarifications and additional results have addressed most of the reviewer's concerns. Below, we would like to offer further clarifications on UniGuide's superiority and distinct advantages over alternative guidance methods that control generation during inference:
> > > >
> > > > &nbsp;
> > > >
> > > > > Highlight the difference to other controlling techniques
> > > >
> > > > - **UniGuide maps the condition to the configuration space**:
> > > >   - Prior works [1,2,3], centred around classifier-guidance [1],  guide towards conditions that describe global input properties. For example, [2,3] control the generation of molecules given a desired quantum property. We highlight that such prior works are limited in their applications and cannot (directly) condition on geometric conditions as introduced in our work.
> > > >   - [1,2,3] control the generation during inference usually by mapping the output of the diffusion model to the condition's space $(Z \to S)$, by using a classifier/regressor $f_{\theta}(z_t)$. In other words, the distribution $p_{\theta}(c |z_t)$ takes the form $p_{\theta}(c|z_t) =  \mathcal{N}(c |f_{\theta}(z_t), I).$ In contrast, UniGuide maps the source condition to the configuration space $(S \to Z)$ which simplifies the gradient signal computation  $\nabla_{z_t} p_{\theta}(c |z_t)$ to a squared error between the sample $z_t$ and the target condition $c = C({s}, \hat{z}_t)$ and eliminates the need for additional networks, see Eq.15.
> > > > - **UniGuide is a self-guiding method**: UniGuide modifies the reverse process of the unconditional diffusion model without relying on additional networks to guide the generation, unlike [1,2,3] that use additional differentiable networks for guidance. This highlights the flexibility of UniGuide as it does not require additional training: neither training a specialised conditional diffusion model nor training additional classifiers/regressors for guidance.
> > > > - **UniGuide is a gradient-based method**: In our experimental evaluations, we include other controlling techniques during inference, such as inpainting or position guidance.
> > > >   - For SBDD, inpainting directly inpaints the noised condition to the generation, leading to a perturbed denoising process through the used hard constraints.
> > > >   - For LBDD, ShapeMol+g directly leverages the positions of the reference ligands and adds additional position corrections to the conformations based on distance measures.
> > > >
> > > > &nbsp;
> > > >
> > > > > Elaborate on the superiority of UniGuide
> > > >
> > > > - **Unification** through the condition map: UniGuide leverages a novel condition map to incorporate diverse geometric conditions, such as protein structures, molecular fragments, and molecular surfaces. This extends unconditional base models to diverse drug discovery tasks. Beyond established drug discovery tasks, UniGuide enables tackling entirely novel tasks where neither baselines nor sufficient data exists, e.g. the tasks of density-based drug design or symmetric proteins. We kindly refer the reviewer to the general response and the figures of the PDF.
> > > > - **Equivariance**: Different from the text and image domains, equivariance is an important inductive bias of molecular generative modelling. In Theorem 4.1, we derive the conditions on UniGuide's condition map that enable guidance while respecting the molecular domain by maintaining equivariant updates.
> > > > - Empirically, our results show that through UniGuide's soft gradient-based updates, we achieve state-of-the-art performance by providing the best trade-off between how well we incorporate the condition into the generative process (LBDD: shape similarity and graph similarity,  SBDD:  Vina) and the overall quality of the molecules (QED, SA, validity). We kindly point the reviewer to Tables 2 & 3 in the PDF.
> > > > - Finally, we believe our follow-up response shows UniGuide's superiority over validity guidance. However, we are unsure if the reviewer has seen it. We provide results demonstrating that our condition map is more reliable and outperforms validity guidance regarding shape similarity, highlighting its effectiveness.
> > > >
> > > > &nbsp;
> > > >
> > > > In summary, like classifier guidance, UniGuide controls the generation without additional training. In addition, UniGuide extends this approach [1] to the molecular domain by mapping the source condition to the configuration space. This enables direct self-guidance without additional networks while preserving equivariance. Our experimental evaluations show that UniGuide outperforms various baselines, including other techniques that control the generation solely during inference across multiple downstream tasks. We are thankful for the reviewer’s feedback and hope that our answers sufficiently clarify the remaining concern.
> > > >
> > > > &nbsp;
> > > >
> > > > [1] Dhariwal et al. Diffusion models beat gans on image synthesis
> > > >
> > > > [2] Equivariant Energy-Guided SDE for Inverse Molecular Design
> > > >
> > > > [3] Training-free Multi-objective Diffusion Model for 3D Molecule Generation

---

> > > > > ### Comment · Reviewer_ZcJd · 2024-08-14
> > > > >
> > > > > Thank you for the follow-up clarification. I have seen the new experiments about comparing UniGuide and validity guidance. However, I'm curious about what the specific difference is in terms of implementing these two methods? From my understanding, they both use the loss in the coordinate space derived from the geometric constraint to guide the diffusion sampling process.

---

> > > > > > ### Author Response · Authors · 2024-08-14
> > > > > > **Technical difference between UniGuide and validity guidance**
> > > > > >
> > > > > > We thank the reviewer for engaging in the discussion and raising this important clarifying question. It is correct that both UniGuide and the validity guidance from [1] compute the loss in coordinate space.
> > > > > >
> > > > > > However, UniGuide's condition map explicitly maps the condition to target points related to the surface, expressed as $c = C(s, \hat{x}_t)$, where $s$ represents the surface and $\hat{x}_t = f\_\theta(x_t, t)$ is the clean data conformation approximation. In contrast, [1] leverages the shape descriptor $S(x_t)$ to decide if a point $x_t$ is inside the volume but then does not use the surface for the gradient update but regresses against all surface-generating points.
> > > > > >
> > > > > > Note that [1] relies on this differentiable function $S(x)$ to implicitly define a surface via $S(x)-\gamma=0$. For a point $x_t$ that satisfies $S(x_t)<\gamma$, the gradient computes as follows:
> > > > > >
> > > > > > $$
> > > > > > \begin{aligned}
> > > > > > \nabla_{x_t} S(x_t) = \nabla_{x_t} \Big [ -\sigma \log \Big ( \sum_i^N \exp(-\|x_t-y_i \|_2^2 / \sigma ) \Big ) \Big ] \\
> > > > > > \end{aligned}
> > > > > > $$
> > > > > >
> > > > > > $$
> > > > > > \begin{aligned}
> > > > > > \qquad \qquad & = \frac{1}{\sum \exp(\dots)} \sum\_i^N \underbrace{\exp(\dots)}\_{\omega_i} \nabla\_{x_t} \|x_t-y_i \|_2^2  \\
> > > > > > \end{aligned}
> > > > > > $$
> > > > > >
> > > > > > $$
> > > > > > \begin{aligned}
> > > > > > \qquad \qquad & = \frac{1}{\sum \omega_i} \nabla\_{x_t} \sum_i^N \omega_i \|x_t-y_i \|_2^2
> > > > > > \end{aligned}
> > > > > > $$
> > > > > >
> > > > > > This gradient formulation is quite similar (up to the weighting) to UniGuide's special case $S=Z$, as it computes an L2 loss on a given conformation ($\{y_i\}$) and does not generalise to arbitrary geometric conditions. In practice, this means that [1] calculates gradients with respect to the surface-defining points of the noisy, generated ligand rather than the condition surface.
> > > > > >
> > > > > > We consider this suboptimal and have experienced that this formulation can lead to instabilities, as it heavily depends on the quality of the noisy samples and never explicitly defines the surface. (Note that we also had similar issues when guiding the clean data point's conformation $\hat{x}_t$ instead of $x_t$.)
> > > > > >
> > > > > >
> > > > > > Within UniGuide, in contrast, the surface is explicitly defined, and the gradient updates relate directly to the surface. UniGuide's condition map provides guidance targets $c = C(s, \hat{x}_t)$ (see Eq. 20 and 21, manuscript) that are directly associated with the surface, either on the surface or projections into it. For these target conditions $c$, we compute $\nabla\_{x_t} \| \hat{x}_t - c \|_2^2$. This technical difference comes with several advantages:
> > > > > > - The separation of the condition from the generated configuration, combined with the subsequent comparison via a direct L2 loss, shifts the focus to appropriately defining the condition.
> > > > > > - UniGuide does not require the condition definition to be differentiable, offering greater flexibility.
> > > > > > - The condition's explicit nature and the more flexible gradient definition enable us to derive various projections, such as setting a margin for projecting into the surface. This precision is challenging to achieve with a distance-based gradient formulation.
> > > > > >
> > > > > > Finally, we would like to reemphasise that UniGuide's formulation is versatile and extends to other tasks, including density-based drug discovery (as noted in our response to D4nw). In contrast, validity guidance shares limitations similar to UniGuide's special case where $S = Z$. We hope this explanation provides a clear and satisfactory understanding of the distinction between how UniGuide implements conditions compared to validity guidance.
> > > > > >
> > > > > > [1] DECOMPDIFF: Diffusion Models with Decomposed Priors
> > > > > > for Structure-Based Drug Design

---

### Official Review · Reviewer_cfg3 · 2024-07-15

**Soundness:** 3
**Presentation:** 3
**Contribution:** 2
**Rating:** 5
**Confidence:** 2

**Summary:**

The paper introduces UniGuide which is a general framework for conducting conditioning over the unconditional molecule diffusion models during inference. To achieve this, Uniguide introduced a concept called a condition map for different applications. With a condition map,it could control the score function by adding a task-related gradient term for generating samples with desired properties. The experiment has been conducted over 3 settings of SBDD, FBDD, LBDD to demonstrate the effectiveness of the proposed method.

**Strengths:**

1. The paper is generally well-written. The motivation is clear and important to the broad audience for taking usage of unconditional models to different complex scenarios.

2. The paper utilizes extensive experiment settings. I appreciate the efforts to conduct experiments in all the SBDD, FBDD, and LBDD settings.

**Weaknesses:**

1. However, the paper's contribution is hard to evaluate. Though the author claims a new general framework for Uniguide, the form takes exactly as a gradient guidance form for diffusion models which has been explored in the previous literature and following ups [1]. Hence, the key contribution, given in the previous works,  limits to a direct application of gradient-based guidances of molecule diffusion to different scenarios. I would like the author to clarify more about the contribution.

2. The important related works are missing such as [1,2]. I suggest the authors do a comprehensive review over the relevant literature.

[1] Equivariant Energy-Guided SDE for Inverse Molecular Design. ICLR 2023
[2]Training-free Multi-objective Diffusion Model for 3D Molecule Generation ICLR 2024

**Questions:**

refer to above

---

> ### Author Rebuttal · Authors · 2024-08-07
>
> We would like to thank the reviewer for their valuable feedback and will address the raised concerns in our answer below.
>
> &nbsp;
>
>  > However, the paper's contribution is hard to evaluate. Though the author claims a new general framework for Uniguide, the form takes exactly as a gradient guidance form for diffusion models which has been explored in the previous literature and following ups [1]
>
> - We acknowledge that UniGuide builds upon the existing framework of gradient-based guidance for diffusion models. However, our primary goal was to **develop a method capable of conditioning on general geometric information, a capability absent in previous work**. We achieve this through two key innovations: self-guidance and the condition map.
>
> Key Differentiators:
> - Condition Map: **UniGuide leverages a novel condition map to incorporate diverse geometric information**, such as protein structures, molecular fragments, and molecular surfaces, as guidance conditions. This is a significant departure from previous work like [1, 2, 3], which are limited to conditioning only on specific (quantum) properties.
> - Self-guidance: UniGuide is a self-guiding approach that modifies the reverse process of the diffusion model without relying on additional networks to guide the generation, similar to prior work [5]. This highlights the flexibility of UniGuide as it does not require additional training (neither training a specialised conditional diffusion model nor training additional networks for guidance). This is unlike prior work [1] that requires training additional regressors to guide the generation.
> - Property-Based Conditioning: In principle, UniGuide can also perform (quantum) property-controlled generation. As this task focuses on global graph properties, it would require additional regressors.
>      - EEGSDE [1], on the one hand, computes the guidance loss between a condition property value $\mathbf{c}$ and the output of a property prediction network that is finetuned over different noise levels of the diffusion process $\mathbf{m}_{\theta}(\mathbf{z}_t,t)$.
>      - In contrast, UniGuide uses a property regressor trained on clean samples, leveraging a clean approximation ( $\mathbf{\hat{z}}_t$ ) of the noisy data ( $\mathbf{z}_t$ ) for guidance. This approach, explored in [2], differs from our focus on self-guidance. While previous studies [1,2,3] explore property-based conditioning with various conditioning mechanisms, our work investigates self-guidance without relying on external regressors or classifiers, and its broad application across various drug discovery tasks.
>
> - Multiple Conditioning: We further indicate that, unlike prior works [1,2], **UniGuide can support conditioning both on geometric conditions (scaffolds, proteins or shapes) and global graph properties (quantum properties)**, purely during inference. However, this requires additional property regressors [2].
>
> - Motivation for new drug discovery strategies: Inspired by the performance on the LBDD task, we further motivate the applicability of UniGuide for the generation of molecules given molecular densities, see App.G and our detailed answer to Reviewer D4nw. We further provide a qualitative example in Fig.1 in the attached PDF.
>
> UniGuide is Not a "Direct Application": Consequently, we can confidently state that our work does not merely constitute a "direct application" of gradient-based guidance. **UniGuide introduces fundamental innovations that enable geometric conditioning**, unlocking novel applications and expanding the capabilities of controlled molecule generation.
>
> > The important related works are missing
> - The reason why we omitted the discussion on property-based conditioning is that the current self-guidance formulation of UniGuide does not support drug discovery tasks involving global graph properties. However, it is possible to combine UniGuide with classifier-guidance to enhance the generated molecules' quality for different downstream drug discovery applications.
> - We acknowledge that the omission of this discussion may have led to some confusion. We will incorporate the suggested prior works [1, 2] into the related works section and describe how they differ from UniGuide.
>
> &nbsp;
>
> We hope that the we were able to appropriately adress reviewer's concerns and are looking forward to a constructive discussion.
>
> &nbsp;
>
> [1] Bao et al. Equivariant energy-guided sde for inverse molecular design
>
> [2] Han et al. Training-free Multi-objective Diffusion Model for 3D Molecule Generation
>
> [3] Hoogeboom et al. Equivariant diffusion for molecule generation in 3d
>
> [4] Dhariwal et al. Diffusion models beat gans on image synthesis
>
> [5] Song et al. Loss-guided diffusion models for plug-and-play controllable generation

---

> > ### Comment · Reviewer_cfg3 · 2024-08-14
> > **Thanks for your response**
> >
> > I carefully check the rebuttal.
> >
> > I am now convinced that Uniguide holds a difference with the direct application of gradient-guided generation and I also appreciate the novelty of geometric condition maps.
> >
> > I increased my scores to appreciate the efforts.

---

> > > ### Author Response · Authors · 2024-08-14
> > > **Thank you for your response**
> > >
> > > Thank you very much for the kind words; we are grateful for an increased score and are excited about the positive feedback on the novelty of the UniGuide framework. We are pleased to hear that you found our additional clarifications on gradient-guided generation satisfactory, which will be included in the camera-ready version of the paper.
> > >
> > > Best regards,
> > >
> > > The Authors

---

### Official Review · Reviewer_YdVq · 2024-07-22

**Soundness:** 3
**Presentation:** 3
**Contribution:** 3
**Rating:** 6
**Confidence:** 2

**Summary:**

This paper presents a framework for geometric guidance of diffusion models to enable flexible generation for protein and small molecule tasks. Their method is based on self-guidance from geometric conditions. They propose a condition map to map geometric conditions to the latent condition space to guide diffusion. They demonstrate experiments on three main tasks: ligand-based, structure-based and fragment-based drug design.

**Strengths:**

The paper addresses a relevant problem in drug design effectively. There is a need for a general framework for guiding diffusion models for the multitude of protein and small molecule design tasks. This work proposes a novel solution by focusing on geometry-based conditioning, and using a condition map to map different conditions to a common space. They provide extensive experimental evaluation on three major tasks, achieving comparable performance, or outperforming recent methods.

**Weaknesses:**

A key benefit of this approach is the generalizability of the method. The main concerns I have are in relation to this. Regarding training the model, can the authors provide additional details on the training? For example, are all three tasks trained with different conditioning to guide the model, together? Or is one model trained separately for each task? What is the effect of training on multiple tasks together? For example, if one model is trained per task, then it is difficult to see the benefit of this approach, other than in the conditioning map.

In the baseline experiments, I feel that there are some related work missing on works that condition diffusion models. For example, how would a simple conditioning mechanism, such as the ones used in the image-text generation literature, fare against this approach? How would this method compare to other latent diffusion models, such as [1-2]? It seems there are a multitude of latent diffusion models for generation out there, I find it difficult to compare those with this. The related work section also does not contrast this work with others in great detail.

### References
[1] McPartlon, Matt, et al. "LATENTDOCK: Protein-Protein Docking with Latent Diffusion."
[2] Watson, Joseph L., et al. "De novo design of protein structure and function with RFdiffusion." Nature 620.7976 (2023): 1089-1100.

**Questions:**

- What is the relation with this work and latent diffusion? Could one similarly encode the geometric conditions using a latent encoder?
- What is the computational complexity in training these models?
- What other design tasks could this extend to?

**Limitations:**

Yes

---

> ### Author Rebuttal · Authors · 2024-08-07
>
> We thank the reviewer for the feedback and positive evaluation of our method! We provide detailed answers in the following.
>
> &nbsp;
>
> > Additional details on training required for UniGuide’s generality
> - We highlight that **UniGuide does not require extra training** as it controls the generation of pretrained unconditional diffusion models during inference using guidance.
> - Therefore, Uniguide can be directly applied to a suitable (unconditional) model, i.e. a model that matches the configuration space required for the task and is trained on a fitting dataset (e.g. ZINC). We discussed this in the limitations, but we will ensure that this aspect is clearly conveyed in the updated manuscript.
> - Drug design tasks have varying requirements. SBDD and FBDD need protein information, while LBDD and linker design do not. Therefore, the underlying model to which we apply UniGuide varies with the required configuration: We use the unconditional protein-ligand model from [1] for SBDD and FBDD and the molecular generative models ShapeMol or EDM for LBDD and linker design.
> - In summary, UniGuide’s benefit is additive: It does not require a specialised model (no extra training), and the same model can be applied to multiple tasks.
>
> > Comparison with simpler conditioning mechanisms?
> - Controlling the generation of diffusion models using guidance in combination with additional classifiers/regressors is common for the image and text domains [7]. In contrast, UniGuide is a self-guiding [6] approach (Eq.12) that does not require additional models to control the generation, unlike [7]. Another key difference is our **focus on the molecular domain, enabling unified guidance while respecting the requirements of molecular structures: (1) geometric conditions and (2) equivariance**.
> - The condition map is the central element in achieving this. It maps from the space of source conditions S to the configuration space Z (the diffusion model’s output space), realising (1), and maintaining equivariant updates (2) when satisfying Theorem 4.1.
> - We compare to (i) conditionally trained diffusion models and (ii) an inpainting-inspired [1] technique for SBDD and FBDD and show that UniGuide is generally favourable, see e.g. Tab.2.
> - For LBDD, UniGuide is the first method to successfully tackle this task purely at inference time using the surface condition map.
>
> > Q: Computational complexity?
> - Inference:
>     - The computation cost associated with UniGuide results from computing the guidance gradients at inference time, see Eq.15.
>  We report runtime comparisons in App.E.2. UniGuide is slightly slower than the inpainting-inspired method [1]. At the same time, the DiffSBDD-cond model provides a baseline for pure sampling without guidance as the computational cost for controlled generation was already invested upfront for the conditional training (which is much higher).
> - Training:
>     - Details on the unconditional training of EDM and ShapeMol[U] are provided in App.C and App.D.1, respectively. For the SBDD experiments, we use DiffSBDD checkpoints, as provided by the authors [1]; see App.E.2.
>
> > Q:Relation with latent diffusion models? Could one encode geometric conditions using a latent encoder?
> - UniGuide can readily be applied to guide a latent diffusion model, for example, GeoLDM [4] instead of EDM [5].
> - In the special case when the source condition is a molecular structure ($S=Z$), one could use the encoder of the latent diffusion model to map the condition to the latent space and guide with UniGuide there.
> - In the general case of $S \neq Z$, a possible solution is to leverage the decoder D of the respective latent diffusion model and keep the condition map in the data space. That is, instead of computing $C(s, \hat z_t)$, one estimates the clean latent code and applies the decoder afterwards: $C(s, D(\hat z_t))$.
>
> > Q: Does UniGuide extend to other tasks? How does it compare to [2-3]?
>
> UniGuide also applies to the domains presented in [2-3]. We did not include them in the current version of the manuscript as they focus on docking and protein generation only, while we focus on drug discovery centred around small molecules.
> - However, we checked the mentioned papers and would like to share some ideas and results on how the presented tasks can be accomplished with UniGuide:
>     - In the context of docking [2], UniGuide could be used to condition on information about the protein complex. Contact information, for example, could be leveraged by guiding to specific positions of the C-a atoms. Coarser positional information, cf. Fig.G.1 [2], could be done in a similar fashion to the shape-based generation presented for LBDD.
>     - Compared with the settings from [3], **UniGuide can be used as an alternative to generate symmetric proteins**. We adopted this idea and chose the C8 symmetry for this PoC. In particular, we designed a pie-like volume element that resembles the required angles and guided the protein generation with the surface condition map from Eq.21. We present the results of this experiment in Fig.2 in the pdf.
> - Additionally, as prompted by reviewer D4nw, we expand the exploration of Density-based Drug Design in Fig. 1 of the PDF.
>
> &nbsp;
>
> We hope that our answers sufficiently clarify the raised questions. We are thankful for the reviewer’s suggestion to investigate the protein settings and are excited about our results for generating symmetric proteins.
>
> &nbsp;
>
> [1] Schneuing et al. Structure-based Drug Design with Equivariant Diffusion Models
>
> [2] McPartlon et al. LATENTDOCK: Protein-Protein Docking with Latent Diffusion
>
> [3] Watson et al. De novo design of protein structures and function with RFdiffusion
>
> [4] Xu et al. Geometric latent diffusion models for 3d molecule generation
>
> [5] Hoogeboom et al. Equivariant diffusion for molecule generation in 3d
>
> [6] Song et al. Loss-guided diffusion models for plug-and-play controllable generation
>
> [7] Bansal et al. Universal guidance for diffusion models

---

> > ### Comment · Reviewer_YdVq · 2024-08-13
> >
> > Thank you for the detailed response and improving the clarity of understanding the method.  I believe that the rebuttal adds clarity to the paper. I appreciate the additional results in generating symmetric proteins. However, as other reviewers mentioned, the geometric guidance led to marginal improvements in performance. Further, the additional experiments on rebuttal Table 2 only demonstrate minimal performance improvements. As a result, I maintain my score.

---

> > > ### Author Response · Authors · 2024-08-13
> > > **Thank you and additional clarifications**
> > >
> > > &nbsp;
> > >
> > > Thank you for your response and for recognising the additional results we provided as part of our rebuttal. We are glad that our explanations have enhanced the clarity of our paper. Regarding the mention of marginal performance improvements, we have addressed this in detail in our response to YA7u, offering a broader perspective on UniGuide's potential impact and objectives. We kindly refer you to that response and hope it provides further clarity. Specifically for the FBDD experiments, while we agree that Table 2 shows UniGuide's on-par performance with a task-specific model, we would like to emphasise the significantly improved VINA scores by 0.5, as demonstrated in Table 13 of the manuscript.
> > >
> > > &nbsp;
> > >
> > > We hope these clarifications highlight UniGuide's full potential. Thank you once again for your valuable feedback and the engaging discussion.

---

### Author Rebuttal · Authors · 2024-08-07

We are pleased with the positive feedback on our work, particularly **noting its motivation and its broad and flexible applicability across various drug discovery tasks**.

&nbsp;

We incorporated clarifications and additions to our evaluations -  please refer to the attached PDF and individual responses for detailed information.

Presentation Style
- We revised the tables’ presentation style as shown in the PDF.
- Initial Distinction: Our quantitative comparisons distinguish between non-diffusion and diffusion-based methods.
- Conditioning Focus: Within diffusion-based methods, we discern the conditioning approaches that control the same backbone. We aim to isolate the effect of different conditioning techniques (inpainting, conditional training, and self-guidance via UniGuide).

LBDD (mainly by YA7u)
- We need to clarify that Tab.1 (Tab.3 PDF) contains various pieces of information that were not presented clearly, obscuring UniGuide's performance:
  - The LBDD task is addressed in different ways: by leveraging only the molecular shape (UniGuide, ShapeMol, VS) or utilising both the shape and the reference structure (SQUID, ShapeMol+g). We indicate this difference in difficulty by a ✓ for methods that only use the shape information and a ✗ for the others.
  - Furthermore, **the ratio metric is the most important** as it combines shape and graph similarities. **UniGuide (EDM) outperforms all other approaches**, highlighting that our approach discovers novel (low graph similarity) ligands satisfying a given shape.
  - Why we include both UniGuide (ShapeMol[U]) and UniGuide (EDM):
    1. To isolate the effect of UniGuide and compare it with the shape-conditioning of ShapeMol, we trained ShapeMol without its conditional part (ShapeMol[U]). The improvements of UniGuide (ShapeMol[U]) over ShapeMol illustrate the performance benefits provided by our guidance.
    2. Since UniGuide can be applied to any base model, we asked what happens if we use an off-the-shelf EDM model instead of ShapeMol[U]. The substantial performance improvement over both UniGuide (ShapeMol[U]) and ShapeMol, combined with the simplicity of applying UniGuide, demonstrates the effectiveness and benefits of our method.
  - We will revise the updated manuscript to provide a clearer and more detailed explanation of these aspects, ensuring it is better understood.

- Inclusion of UniGuide (ShapeMol) in LBDD (Tab.3 PDF, suggested by ZcJd):  Our application of UniGuide with the surface condition map (Eq. 21) to the conditional ShapeMol demonstrates that UniGuide provides additional benefits also for conditional models. We report this as UniGuide (ShapeMol) in Tab.3 (PDF).

SBDD (Tab.1 PDF, suggested by YA7u&ZcJd)
- As suggested, we added extra metrics: VinaScore, VinaMin, and VinaDock, cf. [1,2].
- We also include DecompDiff [1] & IPDiff [2] as baselines.
- Due to a discrepancy in the vina metrics (caused by post-processing [3]), we reevaluated UniGuide and the other baselines and report the updated results for Tab.2 (manuscript) in Tab.1 (PDF).
- We draw the following conclusions from the updated results:
  - Our results, when utilising the same model backbone [3], demonstrate that **UniGuide outperforms alternative conditioning mechanisms**, such as inpainting and conditional training.
  - Furthermore, UniGuide surpasses DecompDiff (no drifts) regarding VinaScore and VinaMin. DecompDiff (all drifts) incorporates additional guidance drifts to enhance the binding affinity. We anticipate that integrating these drifts into UniGuide will further improve performance, and we will present the results for these additions once they become available.
  - We believe that the addition of such drift terms is also crucial for an adequate comparison with IPDiff and DecompDiff, which both add reference and pocket priors to improve the binding affinity specifically.
  - Nevertheless, **UniGuide performs very well just by controlling an off-the-shelf unconditionally trained diffusion model**, proving its generality and effectiveness.

Linker Design (Tab.2 PDF, suggested by ZcJd)
- We implement an inpainting-inspired method for Linker Design and include it in our comparisons, see Tab.2. We use EDM and inpaint the condition fragments, following [3], and observe that the inpainting mechanism alone is not sufficient for the task of Linker Design.
- Additionally, we refer to the experiments in App.F.2, where we evaluate UniGuide for the FBDD (+pocket information) task and compare it to the inpainting technique [3], see Tab.13. These experiments include scaffolding, fragment linking, and fragment growing. Tab.13 demonstrates UniGuide’s favourable performance over inpainting across various metrics. An additional qualitative comparison is available in Fig.7.

**Novel tasks beyond Sec.5** (inspired by YdVq&D4nw)
- Conditioning on densities: We elaborate on the scenario motivated in App.G and refer to this novel setting as Density-Based Drug Design. Fig.1 (PDF) shows how UniGuide can be utilised to condition on densities of atom types or aromatic rings. Please refer to our response to D4nw for extra details.
- Extension to proteins: Even though our main focus centres around small molecules, UniGuide can be used to generate symmetric proteins (C8 symmetry for this PoC), cf. [4]. In particular, we designed a pie-like volume element that resembles the desired angles for the monomer and guided the protein generation with the surface condition map from Eq. 21, see Fig.2 (PDF).

&nbsp;

We are grateful for the opportunity to address the reviewers' concerns and are committed to incorporating their feedback to improve our work further.

&nbsp;

[1] Guan et al. DecompDiff: diffusion models with decomposed priors for SBDD

[2] Huang et al. Protein-ligand interaction prior for binding-aware 3d molecule diffusion models

[3] Schneuing et al. Structure-based drug design with EDM

[4] Watson et al. De novo design of protein structures and function with RFdiffusion

---

> ### Author Response · Authors · 2024-08-13
> **Additional Analyses for SBDD and LBDD**
>
> ### Follow-Up on Comparison of UniGuide with validity guidance
>
> &nbsp;
>
> We want to follow up on the “Validity Guidance” discussion (ZcJd), as mentioned in [1]. **We have conducted experiments using the surface loss computation and self-guidance for the LBDD task.** We selected the hyperparameters $\sigma$ and $\gamma$ in the surface loss computation to achieve a high DICE score between the implicitly defined surface ($S=\gamma$) and the meshes UniGuide utilises for LBDD ($\sigma=1$,  $\gamma=2$, DICE > 0.8). Our surface calculations are performed using the Open Drug Discovery Toolkit (ODDT), which assigns specific radii to individual atom types and employs the marching cubes algorithm to generate meshes [2].
>
> We performed several runs around the above hyperparameter specifications. The runs were similar in performance and we report the best result below:
>
> | |$\text{Sim}_S ~(\uparrow)$ | $\text{maxSim}_S~~(\uparrow)$ | $\text{Sim}_G ~(\downarrow)$ | $\text{maxSim}_G ~(\downarrow)$ | **Ratio** $(\uparrow)$ |  Connect. $(\uparrow)$ | Unique. $(\uparrow)$ | Diversity $(\uparrow)$ | QED $(\uparrow)$ |
> | -------- | -------- | -------- | -------- | -------- | -------- | -------- | -------- | -------- | -------- |
> | Validity Guidance     | 0.59     | 0.76 | 0.20 | 0.20 | **2.96** | 97%| 100% | 0.76 | 0.69 |
> | UniGuide (EDM) | 0.74 | 0.86 | 0.21 | 0.20 | **3.53** | 99%| 99% | 0.73 | 0.74 |
>
>
> **Although Validity Guidance for LBDD yields low graph similarity, the shape similarity remains suboptimal compared to UniGuide.** Additionally, we frequently encounter numerical instability for this guidance term, an issue not present with UniGuide’s formulation of LBDD.
>
> We believe this ablation will provide a valuable comparison for the final version of our manuscript and hope we have addressed the reviewer's remaining concerns.
>
> &nbsp;
>
> [1] Guan, J., Zhou, X., Yang, Y., Bao, Y., Peng, J., Ma, J., ... & Gu, Q. (2023). DecompDiff: Diffusion Models with Decomposed Priors for Structure-Based Drug Design. In International Conference on Machine Learning.
>
> [2] Lorensen, William E., and Harvey E. Cline. Marching cubes: A high resolution 3D surface construction algorithm, 1998
>
> &nbsp;
> &nbsp;
>
> ### Follow-Up on SBDD
>
> &nbsp;
>
> We want to follow up on the discussion regarding baselines for SBDD (ZcJd & YA7u), specifically IPDiff [1] and DecompDiff [2]. As previously mentioned, incorporating a form of "Clash Drift" into UniGuide, particularly within the condition map, is straightforward and helps prevent the generated ligand from colliding with the pocket. **We have implemented this adjustment, and our initial results indicate that similar to IPDiff and DecompDiff, it improves the overall VINA scores.** Specifically, UniGuide's VINA score, compared to the results reported in Table 1 of the rebuttal PDF, **improved significantly from −5.074 to -5.81. These initial results are promising and we are confident that adjustments of UniGuide's guidance parameters will lead to further improvments.**
>
> We again appreciate the reviewers' suggestions to include [1, 2] and believe that these additional results underscore UniGuide's flexibility to readily incorporate various conditions, enabling the comparison of underlying models across different levels. We hope that these findings address all the reviewers' remaining concerns, and we are looking forward to discussing these additional results.
>
> &nbsp;
>
> [1] Huang et al. Protein-ligand interaction prior for binding-aware 3d molecule diffusion models
>
> [2] Guan et al. DecompDiff: diffusion models with decomposed priors for SBDD

---

### Decision · Program_Chairs · 2024-09-25

**Decision:**

Accept (poster)

**Comment:**

The authors present a new approach for imposing geometrical conditions when sampling diffusion models. The authors illustrate their approach on several tasks relevant to drug design and protein engineering. Reviewers raised several points regarding the presentation and requested new experiments. Many of these were addressed well by the authors during the rebuttal, and they promised to include further results in a camera-ready version. Overall, the paper is interesting and potentially valuable if it is as easy and broadly applicable as noted by the authors. Some reviewers noted that the performance was similar to baselines, suggesting a limited utility of the method. However, the authors have clarified their results and experiments, indicating their approach improves results substantially. My primary personal issue with this paper was the unclear technical presentation, albeit unnecessarily verbose, raising my doubts about its technical rigor.